# Early skin seeding regulatory T cells modulate PPARγ-dependent skin pigmentation

Inchul Cho [1,2], Hafsah Aziz [1,2,6], Jessie Z. Xu [1,2,6], Prudence PokWai Lui[1,2], Monica Sen[1,2], Boyu Xie [3], Pei-Hsun Tsai[1,2], Jie Ting Wang[1,2], Hee-Yeon Jeon[4], Jinwook Choi[5], Shahnawaz Ali [1,2] & Niwa Ali [1,2] ✉

The maintenance of adult tissue homeostasis is dependent on the functional cross-talk between stem cells (SCs) and tissue-resident immune cells. This reciprocal relationship is also essential for tissue organogenesis during early life. The skin harbors a relatively large population of regulatory T cells (Tregs) that accumulate within the first two weeks after birth. A functional role for early skin seeding Tregs (ETregs) during the first week of life is currently unexplored. Here, we show that skin Tregs are detected as early as postnatal day 3 (P3) and enter a dynamic flux of activation marker expression. Punctual ETreg depletion from P6-P8, but not later, resulted in defective hair follicle (HF) melanocyte SC (MeSC) mediated skin pigmentation in juvenile life. Transcriptomic analysis of the whole skin on P9 indicated immediate and pronounced changes in MeSC markers, as well as perturbation of PPARγ target genes in the HF. Accordingly, punctual ETreg depletion, combined with short-term PPARγ antagonization, restored skin pigmentation. Single-cell profiling of P9 skin revealed that PPARγ signalling activity is preferentially diminished in the HF epithelium upon loss of ETregs. Finally, we explored changes in the single-cell transcriptome of the human tissue disorder, vitiligo, characterized by a lack of melanin and consequent skin depigmentation. These analyses showed that the HF cells from lesional vitiligo skin exhibit significant downregulation in PPARγ pathway activation, relative to heathy controls. Overall, ETregs in neonatal skin are critical for sustaining HF PPARγ signaling, which is vital for facilitating MeSC mediated skin pigmentation during postnatal development.

Tissue organogenesis during neonatal life is facilitated by temporarily and spatially coordinated crosstalk between immune cells and stem cells (SCs) in the tissue microenvironment. Barrier sites, such as the skin, harbour relatively large proportions of the CD4 lineage of Foxp3+ regulatory T cells (Tregs), which reside proximal to distinct adnexal structures, such as hair follicle (HFs)[1–3]. The HFs represent an important immune cell niche in skin but also accommodate distinct populations of SCs. These include HFSCs and melanocyte SCs (MeSCs), governing hair shaft production and skin pigmentation, respectively.

An intimate association between skin Tregs and the HF epithelia has been well established in recent years. During development, Tregs seed the skin during the first week of life in conjunction with HF

[1]Peter Gorer Department of Immunobiology, King's College London, London, United Kingdom. [2]Centre for Gene Therapy and Regenerative Medicine, King's College London, London, United Kingdom. [3]Imperial College London, London, United Kingdom. [4]King's College London, London, United Kingdom. [5]Gwangju Institute of Science and Technology, Gwangju, South Korea. [6]These authors contributed equally: Hafsah Aziz, Jessie Z. Xu. ✉e-mail: niwa.ali@kcl.ac.uk

morphogenesis, peaking on postnatal day 13 (P13)[4]. Tregs resident in P8–15 skin are essential for establishing both tolerance to commensal bacterial antigens and suppression of type 2 helper T cell mediated fibrous pathology[5,6]. In adult skin, Tregs expressing the Notch ligand Jagged-1, are required for hair regeneration via promoting HFSC differentiation. This SC regulatory function of Tregs is orchestrated largely independently of canonical immunosuppressive pathways. Conversely, during subacute skin injury, Treg control of inflammatory Th17 responses facilitates HFSC mediated barrier regeneration. While these studies have ascribed distinct and specialised roles for adult Tregs and neonatal Tregs during the second week of life[7], a functional role for the foremost afferent population of skin seeding Tregs during the first week of life has not been explored. Specifically, the existence of a neonatal Treg-SC axis during this specific window of time and its relevance to the homoeostatic development and function of the skin remains unknown.

Here, we demonstrate that neonatal Tregs in P6–P8 skin are highly activated, and transient loss of these cells during this time window, but not later, results in defective melanocyte function in vivo, as evidenced by disrupted skin pigmentation in later life. Notably, the transcriptomic changes in melanocytes precede the inflammatory responses induced by Treg depletion, suggesting Tregs may regulate MeSCs independently of secondary mediators such as CD4+ FoxP3- effector T cells (Teffs) or CD8+ cytotoxic T cells. Furthermore, we observed an immediate dysregulation of transcripts associated with the Peroxisome proliferator-activated receptor (PPAR) signalling pathway. Additionally, we identified changes in PPARγ signalling activity throughout developmental stages of human skin and in the pigment-deficient disease state of vitiligo. We defined neonatal Treg regulation of the PPARγ pathway as a mechanistic link between early

skin development and functional MeSC melanogenic activity for establishing pigmentation of skin. Taken together, our findings revealed that neonatal Tregs during the first week of life are essential for the establishment of SC functions during postnatal development of the skin.

## Results

### Skin Tregs are highly activated on Postnatal day 6

The accumulation of Tregs in skin peaks on postnatal day 13 (P13) of life, through microbiota-dependent seeding from the thymus[6]. At this time-point Tregs display a highly proliferative and activated profile. We sought to perform immune profiling at earlier timepoints, prior to P13, upon first entry of Tregs in skin. We first characterised Treg abundance and phenotypic marker expression in steady-state C57BL/6 neonatal skin and skin-draining lymph node (SDLNs) on P6, P9, P12, and later P28 (juvenile) and P49 (adult) timepoints.

Flow cytometric profiling revealed the presence of skin Tregs at detectable but low numbers on P3 (Fig. 1A, B; full gating strategy is shown in Supplementary Fig. 1). The abundance and proportion of CD4+Foxp3+ Tregs steadily increased and peaked during the second week after birth, which is consistent with previous studies (Fig. 1A, B, red line)[6]. This wave of accumulation in P12 skin was unique to Tregs, as our global analysis of other major skin-resident T cell subsets during this time frame did not display a similar pattern, namely dermal gamma-delta positive T cells (dGDTCs) and dendritic epidermal T cells (DETCs), Foxp3- T effector cells (Teffs), and CD8+ T cells (Supplementary Fig. 2A–F). Of note, Treg percentages on P9 and P12 were comparable to that found in adult skin on P49 (Fig. 1A), corroborating a recent study reporting similar findings[8]. By contrast, the percentage of Tregs in SDLNs was less variable and remained steady throughout the

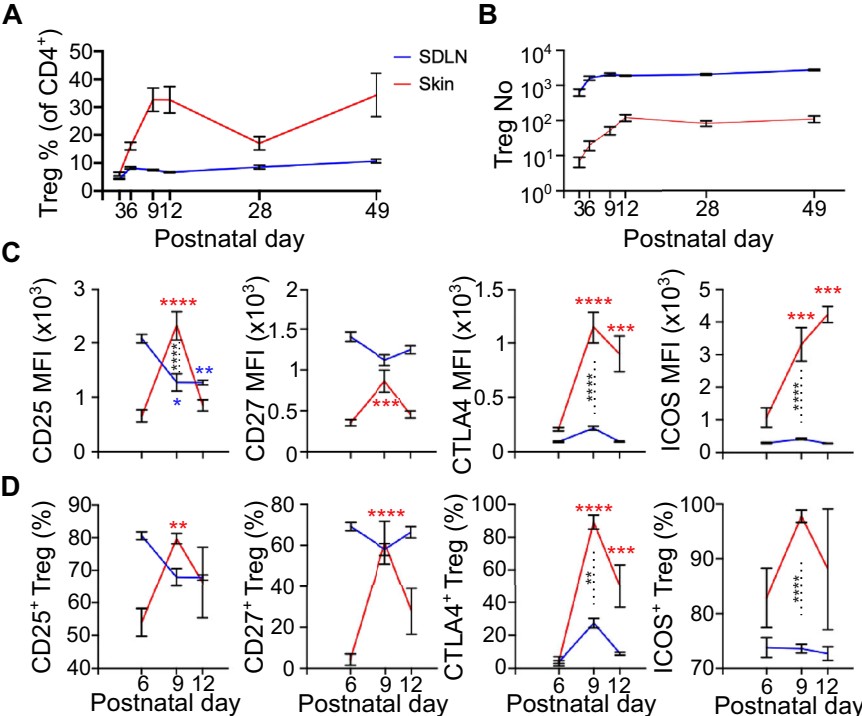

**Fig. 1 | Neonatal skin Tregs accumulate during the first 12 days of life and undergo phenotypic changes. A**–**D** Flow cytometric profiling of skin CD4+ FoxP3+ regulatory T cells (Tregs) (*n* = 4–6 biological replicates) pooled from 2 independent experiments. **A** Percentage of Tregs in the skin (of total CD4+ T cells) in the skin and the skin-draining lymph node (SDLN) on postnatal day 3 (P3), P6, P9, P12, P28 and P49. **B** Absolute number of Tregs in the skin (per 6 cm² of skin) and SDLN (per 10⁵ of total cells) on P6, P9 and P12. **C** Median fluorescence intensities (MFIs) Treg

phenotypic markers, CD25, CD27, ICOS and CTLA4. **D** Percentage of Tregs expressing CD25, CD27, ICOS and CTLA4 (of total Tregs). Graphs show mean ± S.E.M. (*n* = 4–6 biological replicates). *P* value was calculated using One way ANOVA relative to P6. Two way ANOVA with Sidak's multiple comparison test was used to test skin Tregs vs SDLN Tregs. ****p < 0.0001, ***p < 0.001, **p < 0.01, *p < 0.05, ns p > 0.05. Source data are provided as a Source Data file.

analysis period (Fig. 1A, B, blue line). The proliferative index of Teffs, CD8+ T cells, and Tregs was the highest in P6 skin but similar amongst all T cell subsets in the three time-points assessed, as evidenced by equivalent levels of Ki67 expression (Supplementary Fig. 2C). However, in P9 SDLNs, Tregs were more proliferative relative to other T cell subsets (Supplementary Fig. 2F).

Given the rapid influx of Tregs in skin during the early neonatal period of P6-P12, we assessed the expression of the activation-associated markers CD25, CD27, CTLA4, and ICOS, which are preferentially expressed by neonatal Tregs compared to their adult counterparts[6]. Between P6 to P12, the median fluorescence intensity (MFI) values of all markers, except CD27, were significantly higher in P9 skin Tregs relative to their SDLN counterparts (Fig. 1C). In addition, the MFI of all markers were upregulated by at least 2-fold between P6 and P9 skin Tregs, but not in SDLN Tregs (Fig. 1C). The proportions of Treg activation marker expression largely resembled the fluctuations observed in MFI values for both skin and SDLN compartments (Fig. 1D). Interestingly, later skin seeding Tregs (LTregs) in P12 skin expressed lower levels of CD25, CD27, and CTLA4, relative to P9, prompting us to investigate temporal differences in neonatal Treg function. Taken together, these data suggest P3–P12 is an important interval where Tregs accumulate in skin and identifies P6–P9 as a specific time window where early skin seeding Tregs (ETregs) become highly, but transiently, activated in the tissue.

### Neonatal Tregs suppress inflammation and skin pigmentation in later life

Given that Treg activation and proliferation is highly dynamic in P6–P12 skin, we hypothesized that Tregs in this time window may play an important role in regulating either inflammatory responses or postnatal skin development, or both. To test this, we utilized *Foxp3-DTR* transgenic mice where the diphtheria toxin receptor (DTR) is expressed ahead of the *Foxp3* promoter[9]. Administration of DT efficiently depletes Tregs in both SDLNs and skin of *Foxp3-DTR* mice[5,10,11]. To ablate Tregs from P6-P12, we administered DT on P6, P8, P10 and P12 (hereafter referred to as the "ΔTreg" group). We chose P28 as the final timepoint for analysis as the major hallmarks of stem cell (SC) mediated skin development are manifested at this age, namely hair follicle SC (HFSC) activity and melanocyte SC (MeSC) mediated skin pigmentation (Supplementary Fig. 3A).

While Treg sufficient controls on P28 developed normal appearing dorsal skin with black pigmentation, this was markedly impacted in ΔTreg animals, where the skin failed to pigment (Supplementary Fig. 3B). This phenotype was further confirmed histologically using Fontana & Masson (F&M) staining. Melanin granules produced by MeSCs were detected in Treg sufficient controls, but not in ΔTreg skin (Supplementary Fig. 3C). To ascertain whether stem cell regulation is impacted upon Treg depletion, we profiled CD34+ Itga6+ HFSCs and CD117+ MeSCs by flow cytometry (Supplementary Fig. 3D, E). Both the proliferation and abundance of HFSCs were unaffected in ΔTreg skin relative to controls on P28. However, MeSC proliferation was markedly reduced in ΔTreg skin, despite minimal change in their absolute numbers. As such, our data suggest that proliferating melanocytes undergo a turnover[12,13], resulting in a steady population size. Though, it is also possible that differentiated melanocytes in the hair bulb are lost during sample processing for flow cytometry. Overall, we were encouraged to postulate that neonatal Tregs regulate melanogenic function of MeSCs in the HFs during the P6-P12 time frame, in contrast to adult Tregs that control HFSCs activation[10].

To determine if the loss of neonatal Tregs results in systemic inflammation in later life, we firstly monitored body weight gain. Depletion of Tregs during this six-day P6-P12 window resulted in a significant weight reduction by P28 (Supplementary Fig. 3F). This was despite of a repopulating Treg presence that was significantly higher than Treg-sufficient controls (Supplementary Fig. 3I), highlighting the

importance of neonatal Tregs for the maintenance of skin homeostasis. In addition, immune profiling of all major skin-resident T cell subsets revealed an increased abundance and proliferative status of Teffs and CD8+ T cells on P13, only 1 day after the Treg depletion regimen (Supplementary Fig. 3G, H). This inflammatory phenotype persisted even at the later timepoint of P28 (Supplementary Fig. 3I–K). In particular, CD8+ T cells were highly proliferative and outnumbered Tregs, as evidenced by an increased CD8:Treg ratio in ΔTreg mice relative to controls. These data suggest that Treg depletion during this early six-day window leads to an immediate inflammatory response, which persists until later life. Indeed, the same Treg depletion regimen in adult mice does not cause overt skin inflammation[10], supporting the idea that neonatal and adult Tregs are functionally distinct[7]. Overall, the loss of neonatal Tregs from P6–P12 leads to defective MeSC mediated melanogenesis and a consequent defect in skin pigmentation, that may be associated with a systemic inflammatory response.

### Early skin seeding Tregs (ETregs) are required for skin pigmentation

In our initial experiments, we chose a 4-dose DT administration to deplete Tregs from P6 to P12 (Supplementary Fig. 3A). While this resulted in a defective skin pigmentation phenotype, it was also accompanied by reduced weight gain and an immediate inflammatory response that persisted until P28 (Supplementary Fig. 3F–K). Thus, we set out to determine whether defective MeSC function was a result of prolonged systemic inflammation and to also define a precise 'window' of time for Treg requirement. Given ETregs acquired a highly activated profile between P6 to P9, and the less activated profile of later skin seeding Tregs (LTregs) in P12 skin (Fig. 1C, D), we sought to address how transient loss of Tregs at these two timepoints would impact both skin development and local inflammation. To do so, we implemented a 2-dose DT regimen by administering DT on P6 and P8 to deplete ETregs (hereafter referred to as the "ΔETreg" group), and on P10 and P12 to deplete LTregs (hereafter referred to as the "ΔLTreg" group). Efficient Treg depletion in the skin was confirmed one day after the last DT injection on P9 (for the ETreg group) and P13 (for LTreg group) (Supplementary Fig. 4A, B).

To further examine the importance of local Treg recruitment, we also administered FTY720 on P6 and P8 to block lymphocyte egress and inhibit Treg entry into the skin[6] (Fig. 2A). Both the ΔETreg and FTY720-treated groups exhibited reduced dorsal skin pigmentation by P28, mirroring the phenotype observed in the four-dose ΔTreg model (Fig. 2B, C). Quantification of melanin in hair cycle stage-matched follicles revealed a significant reduction in pigment content in ΔETreg and FTY720-treated animals (Fig. 2D, E). As such, local early Treg recruitment in the skin is required for pigmentation. In contrast, ΔLTreg animals displayed normal pigmentation and were indistinguishable from controls (Supplementary Fig. 5A–E). These results support the conclusion that early recruitment of Tregs into the skin is required for proper melanocyte function and pigmentation.

Histologic F&M examination showed a significant reduction in melanin production in ΔETreg, but not in ΔLTreg animals when assessed in hair cycle stage-matched follicles (Fig. 2D, E, Supplementary Fig. 5D, E)[12]. Importantly, Treg ablation during the P6–P8 window in ΔETreg animals showed no significant effect on their ability to gain weight relative to Treg sufficient control groups (Supplementary Fig. 4C), suggesting that early Treg loss does not compromise general health. Furthermore, we observed no significant change in epidermal thickness in ΔETreg skin (Fig. 2F), a typical hallmark of local skin inflammation. These results indicate that transient Treg depletion in early life impairs pigmentation without inducing overt inflammation.

Next, we performed flow cytometry to test the hypothesis that defective skin pigmentation is linked to a disruption of SC maintenance and/or activation. Profiling of MeSCs and HFSCs showed that both the abundance and proportion of Ki67 expression remains

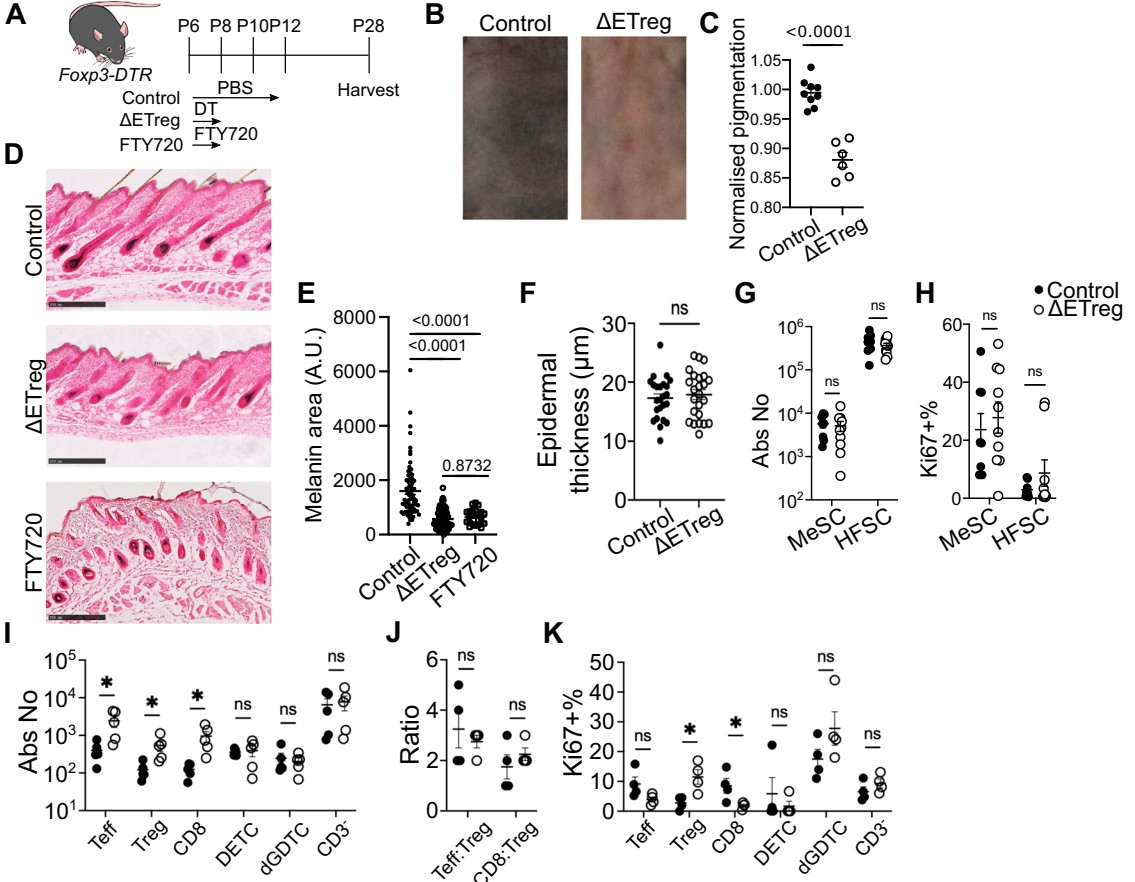

**Fig. 2 | Early neonatal Tregs are required for skin pigmentation. A** Schematic outline of experimental timeline (NIAID Visual & Medical Arts. (10/7/2024). Lab Mouse. NIAID NIH BIOART Source. bioart.niaid.nih.gov/bioart/281). *Foxp3-DTR* transgenic mice received PBS on postnatal day 6 (P6), P8, P10 and P12 (Control), diphtheria toxin (DT) on P6 and P8 (ΔETreg) and FTY720 on P6 and P8 (FTY720). Skin tissues were harvested on P28. **B**, **C** Shaved dorsal skin and quantification of skin pigmentation. **D** Fontana & Masson staining, showing black melanin pigments. Pigmentation was calculated per hair cycle stage-matched follicles. Scale bars represent 250 μm. **E** Quantification of hair cycle stage-matched melanin area per

follicle. **F** Epidermal thickness. Flow cytometric quantification of melanocyte stem cell (MeSC) and hair follicle stem cell (HFSC) (**G**) Abundance and (**H**) Proliferating Ki67⁺ cells. **I**–**K** Flow cytometric characterisation of the skin on P28. **I** Absolute number of skin-resident immune cells. **J** Teff:Treg and CD8:Treg ratio. **K** Proliferation as measured by percentage Ki67 expression. Graphs show mean ± S.E.M. **B**–**H** Data is representative of 4 independent experiments. **I**–**K** Data are representative of two independent experiments. **C**, **F**–**K** Unpaired *t*-test. **E** One Way ANOVA. ****$p < 0.0001$, **$p < 0.01$, *$p < 0.05$, ns $p > 0.05$. Source data are provided as a Source Data file.

comparable between control and ΔETreg groups on P28 (Fig. 2G, H). Therefore, SC maintenance is unaffected in the absence of ETregs.

Given the important role of T cells in disorders of skin pigmentation[14], we assessed whether T cell numbers are impacted in ΔETreg skin on P28. Indeed, the skin T cell pool expanded, but maintained homoeostatic levels of proliferation, Teff:Treg ratios, and CD8:Treg cell ratios (Fig. 2I–K). Additionally, Teff numbers decreased in ΔETreg skin on P9, whilst the abundance and proliferation of other T cell subsets were unchanged (Supplementary Fig. 6A, B). At the molecular level, we observed an increased IFNγ production by CD8⁺ T cells in the skin on P9 following ETreg depletion, but not in other T cell populations (Supplementary Fig. 6C–G), and elevated levels of IFNγ and TNFα in SDLNs (Supplementary Fig. 6H–K). Increased IFN-γ⁺ CD8⁺ T cells are sustained until P17 after ETreg depletion (Supplementary Fig. 7).

To determine if CD8⁺ T cells play a functional role in ETreg-mediated skin pigmentation, we co-depleted Tregs and CD8⁺ T cells using an anti-CD8 monoclonal antibody to determine if MeSC function could be rescued. Under these conditions, skin pigmentation was not restored, suggesting that suppression of CD8-mediated inflammation is not a major mechanism by which ETregs promote MeSC function (Supplementary Fig. 8A). Taken together, we are unable to detect any overt T cell-mediated inflammation in ΔETreg mice,

either immediately on P9 or in later life on P28, that contributes to the defective skin pigmentation phenotype.

## Melanocytes and PPARγ pathway are under the control of ETregs

We next set out to elucidate the cellular and molecular mechanisms responsible for ETreg-mediated control over MeSC function. We were intrigued by the requirement of Tregs during a very short 3-day window from P6-8, and the lack of fulminant inflammation in ΔETreg mice. Supporting our findings are previous studies showing that adult tissue-resident Tregs facilitate tissue homoeostasis largely independently of suppressing conventional inflammatory responses; namely Jagged-1 expressing skin Tregs that promote hair regeneration and Areg-producing Tregs that support lung repair[10,15]. We therefore hypothesized that ETregs regulate either i) the recruitment of specific immune cell subsets we have not assessed through our immune profiling, or ii) specific cellular pathways in non-immune cells, thereby enabling MeSCs to effectively synthesize melanin and maintain postnatal skin pigmentation.

We took a discovery-based approach and reasoned that defining the immediate transcriptomic changes under ETreg control would be key to elucidate the pathways contributing to defective skin pigmentation. We therefore performed bulk RNA-sequencing of whole skin

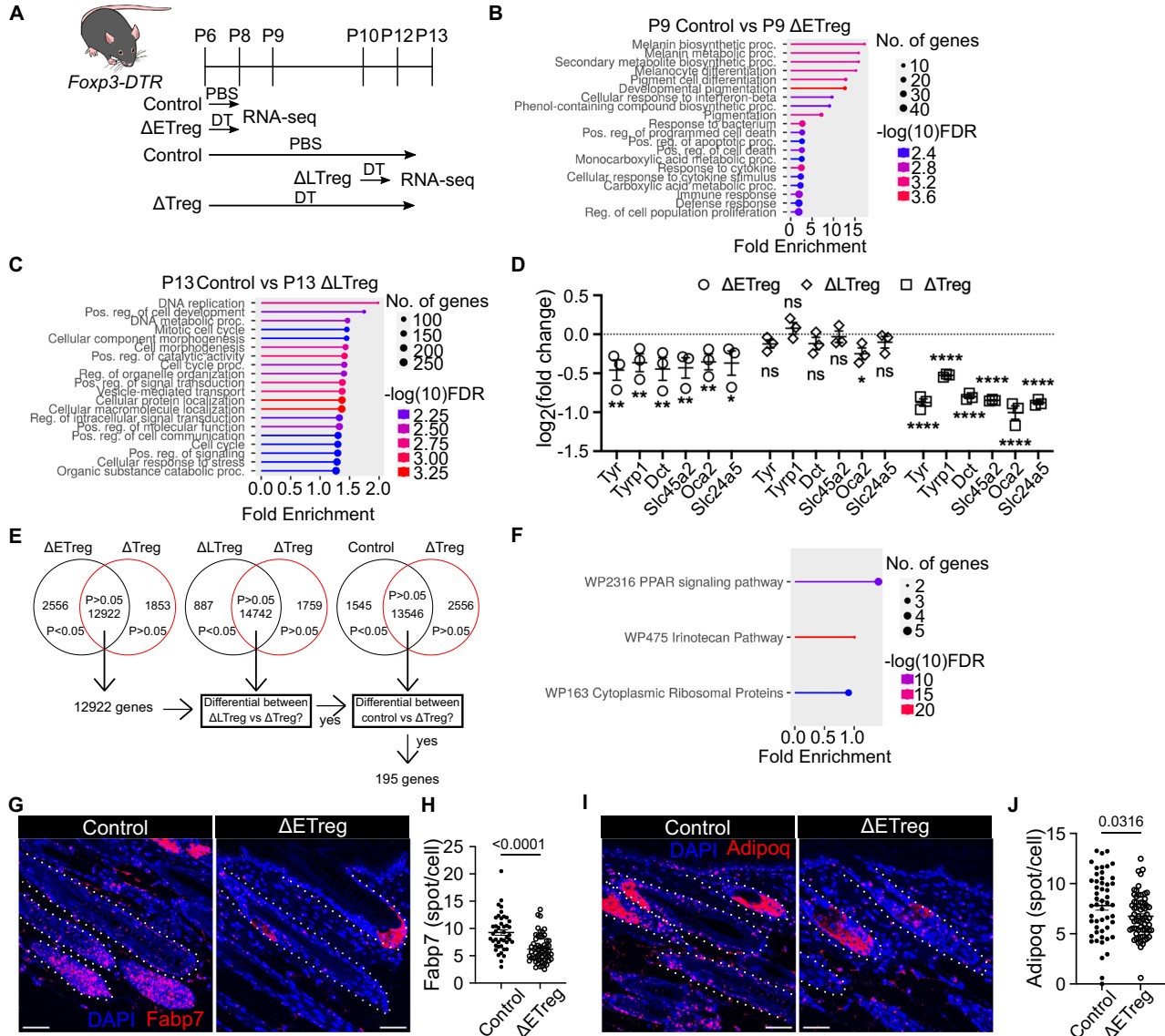

**Fig. 3 | Melanocyte identity and PPARγ activity depend on the presence of early neonatal Tregs. A** Schematic outline of experimental timeline. Skin tissues were harvested for bulk RNA-seq on P9 or on P13 depending on treatment group. (NIAID Visual & Medical Arts. (10/7/2024). Lab Mouse. NIAID NIH BIOART Source. bioart.niaid.nih.gov/bioart/281). **B, C** Enrichment analysis using differentially expressed genes (*p* < 0.05) between age-matched groups. **B** Control vs ΔETreg skin on P9. **C** Control vs ΔLTreg on P13. **D** Normalised expression of melanocyte marker genes. Transcript counts were normalised to age-matched controls (*n* = 2–3 biological replicates). **E** Schematic outline of workflow to identify pigmentation-associated genes. 12922 Transcripts that were non-differentially expressed

(*p* > 0.05) between ΔETreg and ΔTreg were sequentially filtered. Amongst 12922 transcripts, 195 were also differential (*p* < 0.05) between ΔTreg and ΔLTreg, and ΔTreg and control. **F** Pathway analysis using pigmentation-associated genes. **G–J** RNAscope of PPARγ target genes in situ. P9 control and ΔETreg skin. 50 μm scale bar. **G** *Fabp7* and **H** Quantification of *Fabp7* transcripts in the hair follicle (HF). **I** *Adipoq* and **J** Quantification of *Adipoq* transcripts in the HF. **G–J** Data are representative of 3 independent experiments (*n* = 3–4 biological replicates). Unpaired *t*-test. ****p* < 0.0001, ***p* < 0.01, **p* < 0.05, ns *p* > 0.05. Graphs show mean ± S.E.M. Source data are provided as a Source Data file.

taken from Treg-sufficient controls, ΔETreg, ΔLTreg, and ΔTreg groups 24 h after the last DT treatment; P9 for ΔETreg or P13 for ΔLTreg and ΔTreg groups (Fig. 3A). We conducted pathway enrichment analysis using differentially expressed genes (*p* < 0.05) between P9 and P13 skin with ShinyGO[16]. Interestingly, terms such as melanin biosynthesis, melanin metabolic processes, and melanocyte differentiation showed greater than 15-fold enrichment. Additionally, transcriptomic defects in interferon-β responses were evident in the ΔETreg group (Fig. 3B). In comparison, ΔLTreg skin displayed changes primarily associated with cell development and morphogenesis (Fig. 3C). These observations suggest that melanocytes are the major cell lineage impacted upon depletion of ETregs, but not LTregs. This notion is further substantiated by the downregulation of key

melanocyte transcripts (*Tyr*, *Tyrp1*, *Dct*, *Slc45a2*, *Oca2*, *Slc24a5*) specifically in the ΔETreg and ΔTreg groups, but not in the ΔLTreg group (Fig. 3D).

To identify the signalling pathways responsible for pigmentation downstream of neonatal Tregs, we employed specific inclusion criteria based on the observation that pigmentation fails to develop in the ΔETreg and ΔTreg groups. We selected genes that met the following criteria: 1) Not differentially expressed between the ΔETreg and ΔTreg groups, 2) Differentially expressed between the ΔLTreg and ΔTreg groups, and 3) Differentially expressed between the control and ΔTreg groups (Fig. 3E, Supplementary Data 1). We identified 195 genes that fulfilled these criteria. Pathway enrichment analysis revealed dysregulation of transcripts under the control of the peroxisome

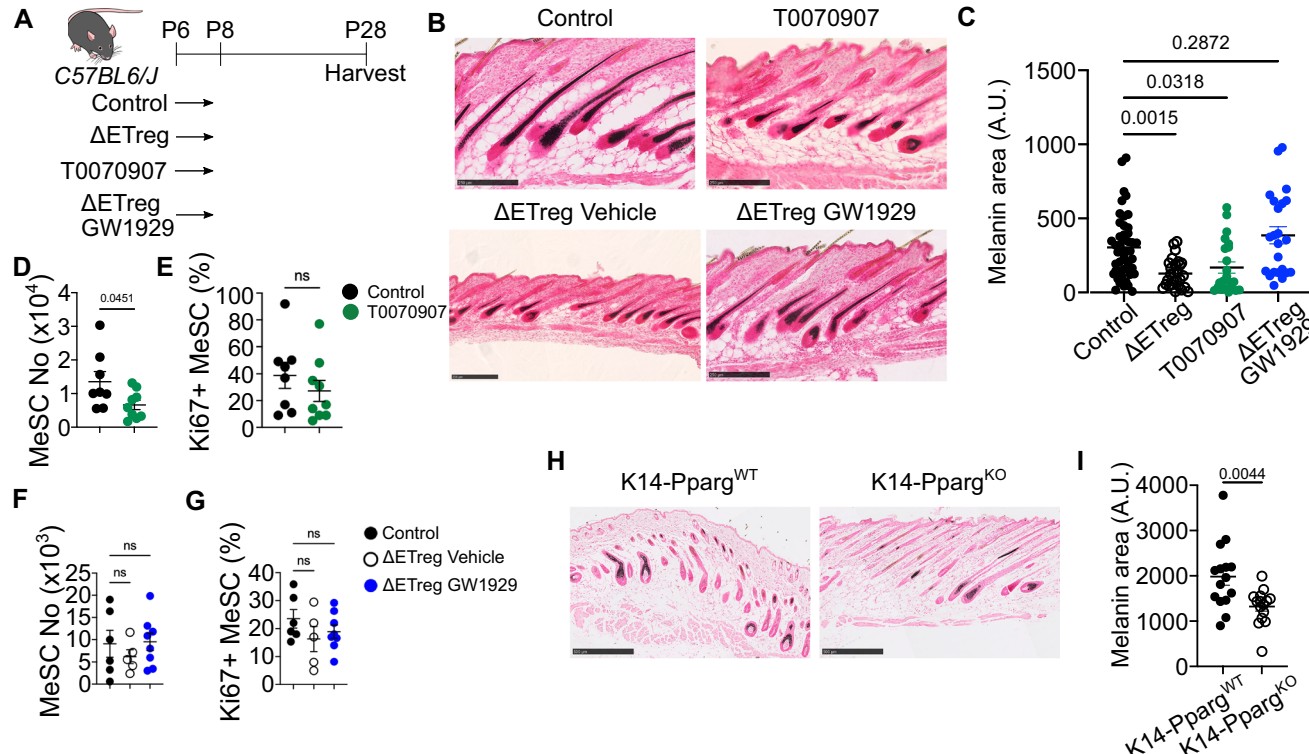

**Fig. 4 | Melanogenic activity requires Treg-PPARγ signalling axis. A** Schematic timeline of PPARγ modulation experiment. Control group received DMSO on P6 and P8. ΔETreg group received DT and DMSO on P6 and P8. T0070907 group received the 2.5 μg/g of PPARγ antagonist, T0070907, on P6 and P8. ΔETreg GW1929 group received DT and 20 μg/g of GW1929 on P6 and P8. (NIAID Visual & Medical Arts. (10/7/2024). Lab Mouse. NIAID NIH BIOART Source. bioart.niaid.nih.gov/bioart/281). **B** Fontana & Masson staining. 250 μm scale bars. **C** Quantification of melanin area in stage-matched hair follicles. **D–G** Flow cytometric characterisation of MeSCs on P28. **D, E** Absolute number and percentage of Ki67+ MeSC upon antagonisation of PPARγ pathway. **F, G** Absolute number and percentage of Ki67+ MeSCs upon agonisation of PPARγ pathway in ETreg-depleted skin (ΔETreg). **H, I** Melanin area in Pparg-sufficient control (K14-Pparg^WT) and upon HF-specific deletion of Pparg (K14-Pparg^KO). 500 μm scale bars. **C, F, G** One way ANOVA. **D, E, I** Unpaired *t*-test. Graphs show mean ± S.E.M. Data are pooled from 2 independent experiments (*n* = 4–6 biological replicates). ***p < 0.001, **p < 0.01, *p < 0.05, ns p > 0.05. Source data are provided as a Source Data file.

proliferator-activated receptor (PPAR) pathway (Fig. 3F). Amongst the three PPAR isoforms, PPARα, PPARβ, and PPARγ, only PPARγ transitions from high expression to low expression from neonatal to adult skin[17]. We therefore focused on validation of PPARγ target genes. To validate the immediate decrease in PPARγ target genes, and to identify potential cell types affected by Treg depletion, we performed RNAscope analysis of neonatal P9 control and ΔETreg skin to quantify the in situ expression of the PPARγ target genes, *Fabp7* and *Adipoq*. The expression of both transcripts specifically in the HFs were significantly attenuated in ΔETreg skin relative to Treg sufficient controls. (Fig. 3G–J), suggesting the possibility that PPARγ activity is affected in cell types residing in the HFs, such as the HF keratinocytes or melanocytes upon depletion of ETregs. Overall, our findings demonstrate that ETreg depletion results in an immediate defect in the skin transcriptome, including reduced PPARγ target gene expression, resulting in a defective melanocyte function later in life. We identified a transient P6-P8 window for establishing a Treg-MeSC axis, during which loss of ETregs leads to significant disruption of the PPARγ signalling pathway.

**PPARγ pathway is necessary and sufficient for pigmentation**
We next sought to determine if the signalling pathways identified in our transcriptomic analysis are responsible for the pigmentation defect downstream of Tregs. In ΔETreg skin, the two main pathways impacted were associated with Type-I interferon response and PPAR signalling (Fig. 3B). These candidate pathways have previously been associated with adult melanocyte function in vitro[18,19], but whether they play a role during postnatal skin development in vivo has not been assessed. To determine if regulation of the interferon pathway is a major mechanism by which ETregs facilitate MeSC functions, we neutralized interferon alpha-receptor (IFNAR) in ETreg depleted mice using an anti-IFNAR monoclonal antibody. Neutralization of IFNAR was unable to reinstate skin pigmentation in ΔETreg mice, ruling out the interferon-β response as pathway mediating the Treg-MeSC axis. (Supplementary Fig. 8B).

The next set of experiments addressed if modulation of the PPAR pathway plays a functional role during steady state neonatal skin development. Amongst three isoforms of PPAR (α, β and γ), PPARγ is expressed at higher levels in neonatal skin relative to adult skin[17]. Therefore, we utilised small molecule modulators targeting the γ isoform, the antagonist T0070907 and agonist GW1929. These small molecules were intraperitoneally injected on P6 and P8 to test the necessity and sufficiency of the PPARγ pathway (Fig. 4A). Inhibition of PPARγ signalling during this early developmental window led to a reduction in melanin production in hair cycle stage-matched follicles, recapitulating the ΔETreg phenotype. Conversely, activation of PPARγ signalling in ETreg-depleted mice restored pigmentation to control levels (Fig. 4B, C). These results demonstrate that PPARγ signalling is both necessary and sufficient to support melanocyte-mediated pigmentation downstream of Tregs.

We then examined whether these changes were associated with altered MeSC dynamics. Inhibition of PPARγ signalling reduced MeSC number but not their proliferation (Fig. 4D, E), whereas PPARγ agonist treatment had no detectable effect on MeSC abundance or proliferation (Fig. 4F, G). This suggests that PPARγ may act through an intermediate cell population to promote melanogenesis, rather than directly supporting MeSC maintenance.

Based on spatial expression of the PPARγ targets Fabp7 and Adipoq in the hair follicle (Fig. 3G–J), and the absence of significant changes in skin-resident T cell numbers or proliferation following PPARγ modulation (Supplementary Fig. 9A–H), we hypothesised that the relevant PPARγ-responsive population is non-immune and epithelial. To directly test this, we generated Krt14-Cre; Pparg-floxed mice (K14-Pparg$^{KO}$) mice to delete PPARγ from the hair follicle epithelium. These mice exhibited markedly reduced melanin levels relative to controls, confirming that epithelial PPARγ activity is required for normal pigmentation (Fig. 4H–I).

Together, these experiments establish a functional role for epithelial-intrinsic PPARγ activity in supporting skin pigmentation during neonatal development. Our findings position PPARγ as a key downstream effector of early Treg activity, linking immune regulation to epithelial cues that govern melanocyte function.

## Epithelial-intrinsic PPARγ signalling activity relies on the presence of ETregs

To begin to understand the diverse cell states in the skin microenvironment, and to further elucidate the cellular mechanisms underlying the ETreg-PPARγ axis, we performed single-cell RNA-sequencing (scRNA-seq) of CD45+ and CD45- cells in the presence and absence of Tregs. We reasoned that since Tregs reside proximal to both HFs and the interfollicular epidermis[10,20,21], early compensatory responses of epithelial resident cell types to ETreg depletion likely precede effects on skin pigmentation in later life. Furthermore, because the expansion of activated self-reactive T cells is observed 3–4 days after Treg ablation[9] we sought to avoid these confounding variables by analysing skin-resident cells 1 day after the last DT treatment, on P9. Also, given skin pigmentation requires the presence of P6-8 ETregs, but not P10-12 LTregs, P9 was a rational timepoint to analyse early transcriptional mechanisms that govern pigmentation downstream of ETregs.

Firstly, we quantified CD45$^+$ immune cell subsets to assess the accumulation of the major inflammatory lymphoid and myeloid cell lineages in skin. In support of our flow cytometric profiling on P9 (Supplementary Fig. 6A, B), pronounced local immune cell activation and inflammation were not observed following ETreg depletion, even though NK cells were moderately increased (Supplementary Fig. 10A, B). We also analysed CD45- immune cells and identified melanocytes (Mel), epithelial cells (Epithelia), hair follicle cells (HF), and other structural cells such as fibroblasts (Fib) and vascular endothelial cells (Vasc). While the overall abundance of these cells was also minimally affected, the proportions of both melanocytes and vascular endothelial cell populations appear to expand in ΔETreg skin (Fig. 5A, B).

Next, we performed a pseudo-bulk differential gene expression analysis to ascertain the magnitude of transcriptomic changes induced across all cell populations in ΔETreg and control mice. The most pronounced changes were identified in the HF and Epithelia clusters encompassing all epidermal keratinocytes of skin, whereas the CD45$^+$ immune cells appear minimally impacted (Supplementary Fig. 10C). When combined, these HF and keratinocytes account for almost 600 differentially expressed genes (DEGs) ($P_{adj} < 0.05$) in the dataset (Fig. 5C and Supplementary Data 2), whereas those of total immune cells amount to approximately 400. Among the hematopoietic CD45$^+$ fraction, the innate lymphoid cell (ILC) cluster was most impacted with 200 DEGs imparted by ETreg depletion (Supplementary Fig. 10C and Supplementary Data 3). T cell numbers changed minimally, despite being considered the main targets of Treg-mediated suppression (Supplementary Fig. 10B). Collectively, these results indicate that the major cellular targets under the control of ETregs in neonatal skin are epithelial and HF cells.

Whole skin bulk RNA-seq and functional in vivo skin pigmentation rescue experiments strongly implicate the PPARγ-pigmentation axis as a major mechanism under Treg control (Figs. 3B–J and 4B, C).

Therefore, we next assessed changes in PPARγ activity across the captured cell types in the presence and absence of ETregs. We selected a set of experimentally validated PPARγ target genes[22] and constructed an activity score using the "AddModuleScore" function built into Seurat package. Increasing scores on this scale indicate increased expression of PPARγ target genes. Within the CD45$^+$ fraction, neutrophils and macrophages displayed the highest activity levels which is in line with the known role for PPARγ in these cell types[23,24]. However, PPARγ scores in these cell types were largely unaffected by ETreg depletion. Instead, amongst the CD45- cells, the most pronounced downregulation of the PPARγ activity score was observed in HF keratinocytes (Fig. 5D, E). This result is in agreement with reduced expression of PPARγ target genes, such as Fabp7 and Adipoq in the HFs (Fig. 3G–J). This is perhaps an unsurprising result given that the HF cluster demonstrated the highest levels of transcriptomic perturbation (Fig. 5C), and the spatial proximity of HF cells to MeSCs in skin. Further analysis of the HF subset showed that 49 of 363 DEGS (~14%) between the control and ETreg depleted groups were known transcriptional targets of PPARγ ($p = 2.25 \times 10^{-9}$) (Fig. 5F). This included genes such as Apoe, Pltp, Abca1, and Igf1, all of which were significantly downregulated in ΔETreg skin (Fig. 5G). To determine whether these transcriptional changes influence melanocyte behaviour, we used CellChat to infer ligand-receptor communication networks[25]. This analysis predicted that IGF1, a PPARγ target expressed in hair follicles, mediates communication between hair follicle cells and melanocytes (Fig. 5H).

To functionally validate this, we cultured Melan A melanocytes in the presence of recombinant IGF1 and observed a significant increase in melanin production (Fig. 5I–J), consistent with its role as a melanogenic factor. Lastly, we investigated how Tregs might modulate PPARγ activity in the hair follicle. Bulk RNA-seq of sorted skin and SDLN Tregs at P9, followed by NicheNet analysis[25], revealed that skin-resident Tregs preferentially express IL10 and TNFSF9, cytokines predicted to regulate Apoe, Pltp, and Abca1 expression in epithelial cells (Supplementary Data 1, Fig. 5K–L). These genes are central to lipid transport and metabolic signalling[26–29], and their downregulation may contribute to impaired PPARγ pathway activation.

Together, these findings demonstrate that neonatal skin Tregs regulate epithelial PPARγ signalling in the hair follicle niche, in part via IL10 and TNFSF9. This PPARγ activity supports expression of melanogenic factors such as IGF1, which may then act on adjacent melanocytes to promote pigment production.

## The PPARγ pathway is implicated in vitiligo and during human skin development

Numerous disorders of skin pigmentation have been described in humans. The most prominent of which is the autoimmune skin disease, vitiligo, in which depigmented skin results from melanocyte destruction[30]. Genome wide association studies have highlighted vitiligo susceptibility loci in genes that support Treg function, including IKZF4 and CTLA4[31,32], Additionally, Tregs are important for restraining skin depigmentation severity in lesional vitiligo skin, further suggesting a functional role for Tregs in disease pathogenesis. Given these associations, we sought to explore whether PPARγ signalling is impaired in pathologically depigmented human skin. To investigate this, we analysed a publicly available scRNA-seq data of human skin samples from 5 healthy donors and 10 vitiligo patients[33]. The final processed datasets yielded 8 main cell types. Consistent with our murine skin single-cell data, we identified two major clusters of keratinocytes in healthy and vitiligo skin; 'HF' characterized by high expression of hair follicle-associated markers KRT14 and KRT15, and 'Epithelia' that expressed the interfollicular epidermal transcript KRT5. We next applied the PPARγ scoring module to assess the level of activity across the captured cell populations[22]. These analyses revealed that only the HF cell type from vitiligo skin exhibited a significant downregulation in PPARγ pathway activation, relative to heathy

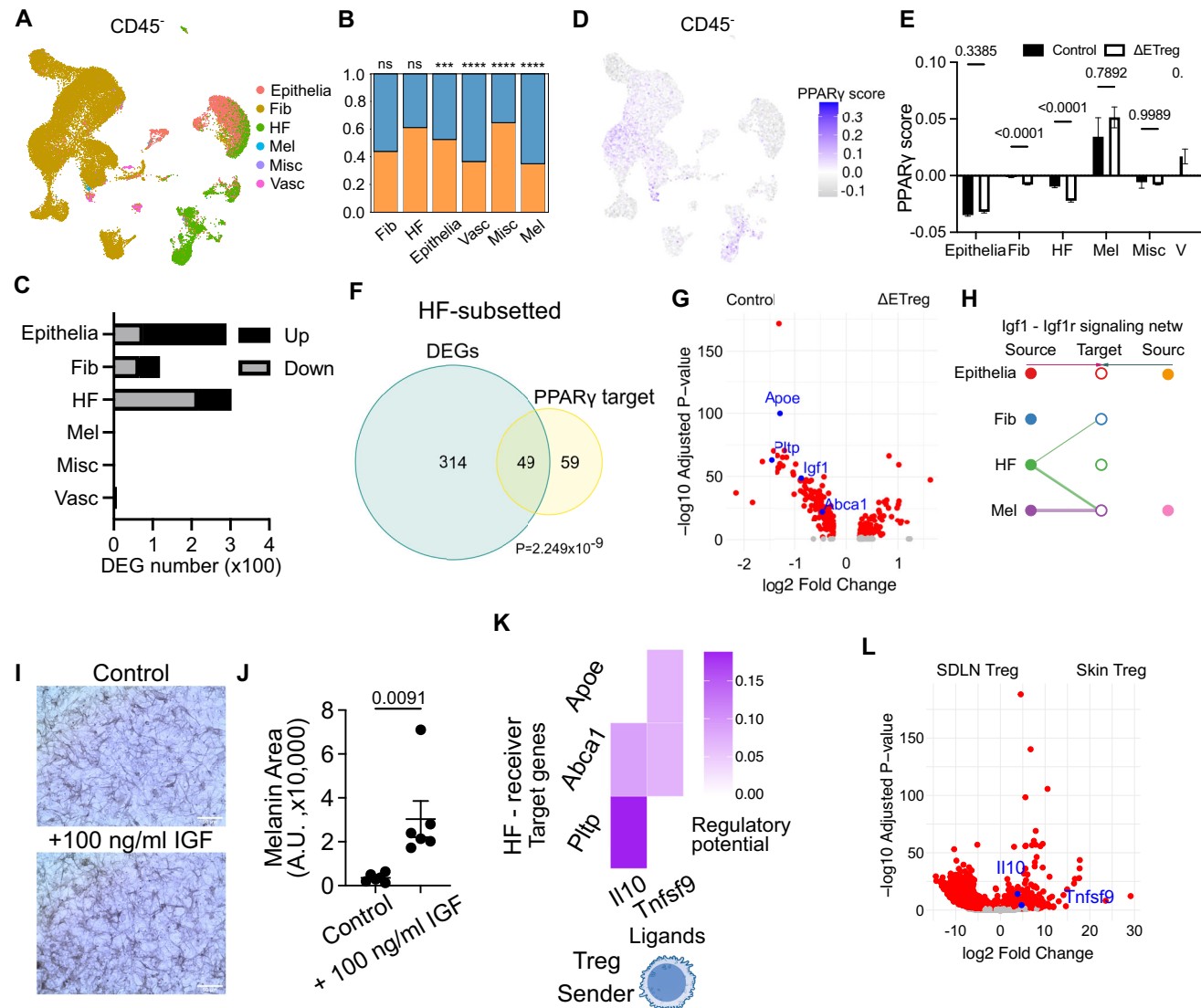

**Fig. 5 | Neonatal Tregs regulate the hair follicle transcriptome and PPARγ activity. A** UMAP representation of CD45⁻ non-immune cells. **B** Ratio of non-immune cells in control and ΔETreg skin. Epithelia epithelial cells, Fib fibroblast, HF hair follicle, Mel melanocyte, Misc miscellaneous, Vasc vascular cell. **C** Number of differentially expressed genes (DEG) by cell type. **D** UMAP representation showing PPARγ score. **E** PPARγ score by cell type between control and ΔETreg skin. *n* = 3 biological replicates per condition. **F** Venn diagram depicting the overlap between the total differentially expressed genes (DEGs) and known PPARγ target genes. P value represents the significance of the overlap as determined by a chi-square test. **G** Volcano plot of representative differentially expressed PPARγ target genes, *Apoe*,

*Pltp, Igf1,* and *Abca1* (blue). **H** Cellchat analysis displaying interactions between cell types (nodes) and the cell-cell interactions (lines). **I** Melan-a cell line cultured in the absence (control) or presence of IGF-1 (100 ng/ml). Scale bar represents 50 μm. **J** Quantification of melanin area. 6 technical replicates per condition. Two-sided unpaired *t*-test. Graphs show mean ± S.E.M. **K** NicheNet analysis using HF cells subsetted from mouse scRNA-seq and sorted Tregs from P9 skin (NIAID Visual & Medical Arts. (10/7/2024). T Cell. NIAID NIH BIOART Source. bioart.niaid.nih.gov/bioart/508). **L** Volcano plot showing higher expression of *Il10* and *Tnfrsf9* in skin Tregs than in SDLN Tregs. Source data are provided as a Source Data file.

controls. (Fig. 6A, B). This result aligns closely to our murine single-cell data where dysregulation of PPARγ signalling was identified in HF cells upon loss of ETregs in neonatal skin (Fig. 5E). Together, these findings raise the possibility that impaired Treg function may disrupt PPARγ signalling in human skin, potentially contributing to disease pathogenesis. Further studies will be required to directly test this hypothesis in humans.

We next set out to address if the PPARγ pathway is involved in human neonatal melanocyte development. In utero, Tregs seed the skin early as gestational week (GW) 18[17,34]. This timeframe aligns closely with murine skin development during the first week of life when ETregs begin to accumulate (Fig. 1). Importantly, this period of Treg infiltration coincides with human HF development, which involves the

migration of melanoblasts to the HF to complete the melanocyte maturation process[35]. Given these associations, we analysed scRNA-Seq data of melanocytes isolated from foetal (9.5–18 GW), neonatal (0 years), and adult (24–81 years) tissue[33,36]. We identified 6 distinct clusters across the dataset. Chi-squared analysis revealed a marked enrichment of adult and neonatal melanocytes for cluster 0, while fetal melanocytes were enriched for cluster 2 (Fig. 6C, D). This observation suggests that melanocyte maturation involves the expansion of cluster 0. Indeed, cluster 0 expresses high levels of mature melanocyte markers such as *DCT, PMEL, TYR*. Cluster 2 expresses high level of proliferation-associated transcripts such as *MKI67, TOP2A and CCNB1*, therefore representing actively dividing melanocytes (Supplementary Data 4). Further analysis identified elevated expression of multiple

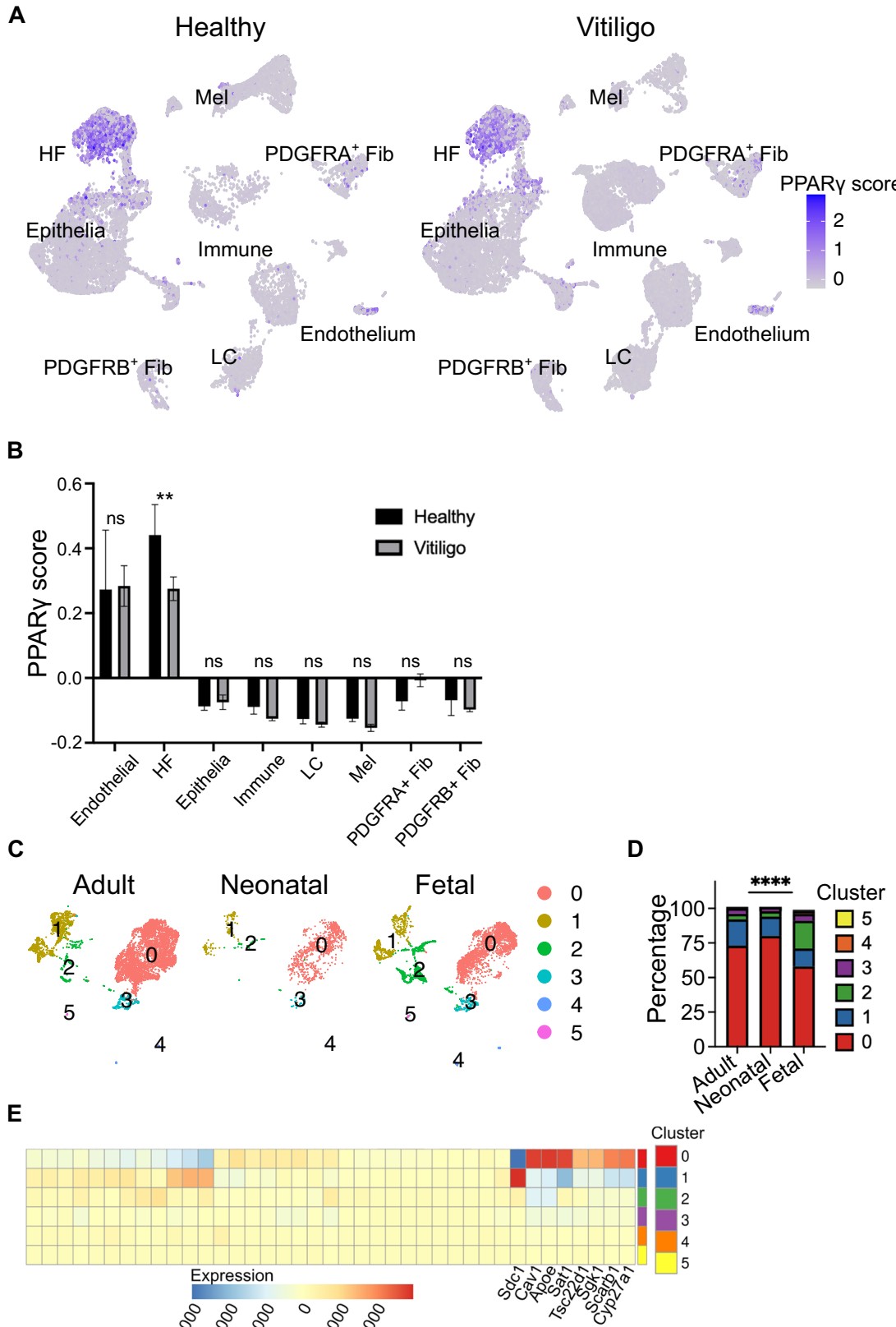

**Fig. 6 | PPARγ target genes are differentially regulated between human developmental stages and diseased states. A**, **B** scRNA-seq analysis of skin from healthy donors and vitiligo patients. **A** UMAP representation of PPARγ target gene expression score. Endothelial endothelial cells, HF hair follicle, IFE interfollicular epidermis, Immune immune cells, LC Langerhans cells, Mel melanocytes, PDGFRA+ Fib PDGFRA+ fibroblasts, PDGFR+ Fib PDGFRB+ fibroblasts. **B** Quantification of PPARγ score. Fisher's LSD test. **C**–**E** scRNA-seq analysis of foetal, neonatal, and adult melanocytes. **C** UMAP representation of clusters. **D** Percentage of cells belonging to individual clusters. Chi-squared test. **E** Heatmap of PPARγ target gene expression. ****$p < 0.0001$, **$p < 0.01$, ns $p > 0.05$. Source data are provided as a Source Data file.

PPARγ target genes in cluster 0 (Fig. 6E)[37], including *CAV1*[38], which regulates melanogenesis and *SCARB1*, which is involved in fatty acid uptake[39]. Overall, an increased PPARγ activity is associated with melanocyte maturation. The transition frow low PPARγ activity to high PPARγ activity temporally coincides with Treg seeding. As such, our findings suggest the possibility that human Tregs may modulate PPARγ activity during human skin development.

Collectively, our analysis of human transcriptomic data suggests that the PPARγ pathway is active during melanocyte development and is preferentially disrupted in HF cells from the pigment-deficient disease state Vitiligo.

## Discussion

Conventionally, Tregs are defined by their robust capacity to dampen tissue inflammatory responses. More recently, there is an increasing realization of the important non-conventional role of adult Tregs in supporting normal tissue maintenance and repair that is either independent of, or mutually codependent with, control of local inflammation[5,10,11,15,20,40,41]. In neonates, Tregs are instrumental in establishing mechanisms of peripheral tolerance to both tissue-specific and commensal bacterial antigens[6,7], as well as regulation of the skin stromal cell niche during the second week of life[5]. Furthermore, adult Tregs support the bone marrow and skin stem cell (SC) niche[10,42,43], but whether a Treg-SC axis exists in neonatal tissue has not been established.

Here, we show that early skin seeding Tregs (ETregs) in the first week of life play a crucial role in establishing tissue homoeostatic processes that manifest in later life. We observed fluctuations of the critical Treg effector molecules CD25 and CTLA4 between postnatal day 6 (P6), P9, and P12 skin, suggesting a temporal dynamic in Treg function during early neonatal stages. Indeed, transient depletion of P6-P8 ETregs, but not P10-P12 late skin seeding Tregs (LTregs), led to impaired melanocyte stem cell (MeSC) mediated skin pigmentation on P28. This indicates that the early neonatal period represents a critical phase for establishing the Treg–MeSC axis and functional maturation of MeSCs.

Interestingly, we also observed a compensatory influx of Tregs following neonatal depletion, resulting in higher Treg numbers in ΔETreg skin compared to controls by P28 (Fig. 2I). A previous study has shown that neonatal and adult Tregs are functionally distinct[7], raising an important question about whether the repopulating Tregs can fully restore the developmental roles of their neonatal counterparts.

To identify the molecular mechanisms responsible for Treg-mediated pigmentation, a whole skin transcriptomic analysis after ETreg depletion identified melanocyte dysfunction and modulation of the PPARγ pathway as under ETreg control. These changes occurred immediately, within 24 h, following ETreg depletion before any signs of a fulminant skin inflammatory response. Restoration of neonatal PPARγ signalling re-instated skin pigmentation, suggesting that this pathway plays a major role downstream of ETregs. By contrast, LTreg depletion leads to transcriptomic changes in genes associated with tissue morphogenesis, suggesting that LTregs may regulate the structural integrity of the skin (Fig. 3C). In line with this observation, depletion of late-seeding Tregs on P8 and P15 leads to the absence of dermal white adipose tissue in later life[5]. We observed similar loss of adipose tissues upon a 4-dose depletion of neonatal Tregs on P6, P8, P10 and P12 (Supplementary Fig. 3C). Together, these observations strongly support temporal dynamic in Treg function. Our flow cytometric profiling of Treg activation markers, such as CD25 and CTLA4, suggest that an intrinsic functional difference between early and late neonatal Tregs are responsible for distinct phenotypes produced by temporal variation in the depletion regimen. Another plausible explanation for the phenotypic differences is that the rapidly developing postnatal skin is only dependent on ETregs for melanogenic activity.

In addition to its immunomodulatory functions, the PPARγ pathway plays a key role in the accumulation and adaptation of Tregs within adipose tissue microenvironments, such as the dermal white adipose underlying the dermis[44]. However, various skin resident T cell populations were unaffected by transient antagonization of the PPARγ pathway in neonatal mice (Supplementary Fig. 8B). This suggests that the Treg-melanocyte axis may operate independently of skin-resident adaptive immune cells. Moreover, skin pigmentation was not restored in ΔETreg mice upon depletion of CD8[+] T cells or blockade of the interferon-β response (Supplementary Fig. 6A, B). These observations support the idea that suppression of inflammation is not the major mechanism by which ETregs promote melanocyte mediated skin pigmentation.

Finally, our whole tissue single-cell analysis suggests that Tregs preferentially regulate PPARγ activity within the HF epithelium (Figs. 3I–L and 5I). Other cell types, such as neutrophils, may contribute to the regulation of PPARγ activity following ETreg depletion. However, these cell types remain unaddressed by our flow cytometric profiling and single-cell RNA-seq analysis, which are unable to capture these populations. Globally, the transcriptome of HFs was impacted most significantly relative to other detected cell types in skin immediately upon depletion of ETregs (Fig. 5E). Similar findings were reported in a recent study assessing the transient loss of Tregs in the contexts of lung cancer and injury induced inflammation. Within 2 days of Treg ablation there were pronounced transcriptional responses in stromal cells, but not adaptive immune cells[45]. The authors demonstrate that this Treg 'connectivity' to non-haematopoietic cell types in lung is facilitated by in situ proximity to these first responding cell types. Given that Tregs reside in proximity to epithelial cells and MeSCs within the HF regions supports the notion that Tregs are highly connected to HFs in skin[20]. We also demonstrated the relevance of this connection in the human skin pigmentation disorder, vitiligo, where Tregs have been implicated in restraining clinical severity. Our analyses have uncovered preferential disruption of PPARγ signalling within the HF epithelium in diseased skin relative to healthy controls (Fig. 6B). Furthermore, we note that PPARγ activity remains low until Treg seeding, and increases after 18 (Fig. 6C–E), when Tregs are known to seed human skin[46]. This raises the possibility that the presence of Tregs in human skin is also required for the activation of PPARγ activity. Together, these observations suggest that Tregs during very early life are functionally poised to modulate PPARγ responses in the HF epithelium and that modulation of this pathway is also impacted in human diseased skin where Tregs play a functional role.

A major limitation of our study is the reliance on pharmacological modulators of PPARγ, rather than genetic models for pathway manipulation in vivo. While neonatal administration of the PPARγ antagonist T0070907 reduces melanocyte abundance in wild-type mice, (Fig. 4D) it does not have the same effect in ΔETreg animals (Fig. 2G). This suggests that distinct mechanisms may underlie pigmentation defects in these two contexts. One possibility is that other melanocyte-regulatory pathways, such as Wnt or TGFβ signalling[47,48], may be impacted upon ETreg depletion. Though, these pathways were not identified as being immediately altered following ETreg depletion. However, our transcriptomic analyses did not identify these pathways as being immediately impacted after Treg loss.

An alternative explanation lies in cell-type sensitivity: we observed that melanocytes display the highest PPARγ activity scores at baseline, potentially rendering them more susceptible to pharmacological inhibition (Fig. 5E). This may account for the differential outcomes observed between ETreg depletion and PPARγ blockade.

Overall, our study provides new insights into the function of the foremost skin-seeding Tregs during neonatal life. Immune profiling during the first 12 days indicates a temporal dynamic in Treg function, whereby early skin seeding Tregs, but not later seeding Tregs,

potentiate PPARγ signalling to facilitate melanocyte function. Pertinently, GWAS studies have implicated Treg malfunction in the human melanocyte-associated disease, vitiligo[49]. Future studies investigating the frequency, phenotype, and, most critically, the spatial co-localization of neonatal skin Tregs with stem cell populations in the hair follicle epithelium at single-cell resolution are warranted. Such efforts may uncover previously unappreciated Treg modalities that shape early-life skin development and contribute to disease susceptibility later in life.

# Methods

## Mouse studies
All animal experiments were subject to local ethical approval and performed under the terms of a UK government Home Office licence (PP6051479). All mice were outbred on a C57BL/6J background. Both male and female mice were used. B6.129(Cg)-Foxp3^{tm3(DTR/GFP)}Ayr/J (Foxp3-DTR) mice were purchased from Jackson laboratories. For Treg depletion, *Foxp3-DTR* mice intraperitoneally received 30 ng/g of DT on P6 and P8. Thymic egress was blocked by intraperitoneal injection of FTY720 (Selleck Chemicals), dissolved in normal saline, at a dose of 10 mg/kg on P6 and P8. PPARγ antagonization was performed by intraperitoneal administration of 2.5 μg/g of T0070907 on P6 and P8. Rescue studies were performed by intraperitoneal administration of 30 ng/g of DT and 20 μg/g of GW1929 on P6 and P8. CD8$^+$ T cells were depleted by intraperitoneal administration of 100 μg of α-CD8 depleting antibody (BioXcell) on P6, P8, P15 and on P22. Interferon-β responses were blocked by intraperitoneal administration of α-Ifnar blocking antibody (Biolegend) on P6 and P8. To assess the role of epithelial-intrinsic PPARγ, we used *Krt14-Cre; Pparg^{fl/fl}* mice, in which *Pparg* is deleted in the keratinocyte lineage. These mice were generated by crossing *Krt14-Cre* mice (Tg(KRT14-cre)1Amc; provided by Abigail Tucker) with *Pparg fl/fl* mice (Pparg^{tm2Rev}; from JAX lab). Pre-weanlings were sacrificed by overdose of pentobarbital. Post-weanlings were sacrificed by CO2 asphyxiation followed by severing a major blood vessel. All efforts were made to minimise suffering for mice.

## Tissue digestion
Single-cell suspensions of full-thickness skin for flow cytometry was performed as previously described[10]. Isolation of cells from axillary, brachial and inguinal lymph nodes for flow cytometry was performed by mashing tissue over 70 μm sterile filters. To prepare dorsal skin cell suspension, shaved mouse skin was de-fatted, minced finely with scissors, and re-suspended in a 3 ml of C10 (RPMI-1640 with L-glutamine with 10% heat-inactivated FBS, 1% penicillin-streptomycin, 1 mM Sodium pyruvate, 1% Hepes, 1× Non-essential amino acid and 60 μM β-mercaptoethanol) supplemented with 2 mg/ml collagenase XI, 0.5 mg/ml hyaluronidase and 0.1 mg/ml DNase in a 50 ml conical. The mixture was digested in a shaking incubator at 37 °C at 255 rpm for 45 min. After a vigorous shaking, the digested mixture was passed through a sterile 100 μm filter fitted onto a 50 ml conical. After pelleting, the filtrate was then filtered once more through a 40 μm strainer fitted onto a new 50 ml conical. Finally, the cells were pelleted once more and re-suspended in 1 ml of C10. Epidermal cell suspensions were prepared by scraping the dermis away with forceps and incubating the layer of epidermis on 3 mL of 0.5% Trypsin-EDTA (ThermoFisher) at 37 °C for 1 h. Epidermal cells were isolated by scraping the trypsinised epidermis on a petri dish containing 4 ml of C10 media. The mixture of cell suspension and scraped skin was filtered through a 70 μm strainer fitted onto a 50 ml conical. Finally, the cells were pelleted and re-suspended in C10. After cell count using nucleocounter (Chememoetec), cells were plated on nunc round bottom 96-well plates (Thermofisher) for staining.

## Flow cytometry
Following isolation from the tissue, cells were labelled stained in PBS for 20 min at 4 °C with a live dead marker (Zombie UV™, Biolegend). Surface staining was performed in brilliant stain buffer (BD Biosciences) for 20 min at 4 °C. For intracellular staining, cells were fixed and permeabilized using reagents and protocol from the Foxp3 staining buffer kit (eBioscience). Fluorophore-conjugated antibodies specific for mouse cell surface antigens and intracellular transcription factors were purchased from eBioscience, BD Biosciences or Biolegend as detailed in the Supplementary Data 1. All samples were run on Fortessa LSRII (BD Biosciences) at the KCL BRC Flow Cytometry Core. Experiments were standardised using SPHERO Rainbow calibration particle, 8 peaks (BD Biosciences, 559123). For compensation, UltraComp eBeads (Thermo Fisher, 01-2222-42) were stained for each surface and intracellular antibody following the same procedure as cell staining. ArC Amine Reactive Compensation Bead Kit (Thermo Fisher, A10346) were used for Zombie UV™ stain. All gating and data analysis were performed using FlowJo v10, while statistics were calculated using GraphPad Prism 9. Strict dead cell and doublet cell exclusion criteria were included for all immune cell analysis, followed by pre-gating for all hematopoietic cells as CD45$^+$. All immune cells were pre-gated as Zombie UV$^-$ CD45$^+$ cells. CD3$^-$ lymphoid/myeloid cells were gated TCRγδ$^-$CD3$^-$ double negative cells. Other lymphoids were gated as TCRγδ$^+$CD3$^+$ dermal γδ T cells (dGDTCs), TCRγδ$^{hi}$CD3$^{hi}$ dendritic epidermal T cells (DETCs), TCDγδ$^-$CD3$^+$CD8$^+$ T cells (CD8), TCRγδ$^-$CD3$^+$CD4$^+$Foxp3$^-$ T effector cells (Teff), and TCRγδ$^-$CD3$^+$CD4$^+$Foxp3$^+$ regulatory T cells (Tregs).

## T cell stimulation
PMA/Ionomycin cocktail (Tonbo Biosciences) was diluted to 1× concentration in C10. Heat-inactivated FBS must be used to make C10 for cell stimulation. 6–8 million live cells were re-suspended in 200 μl of 1× PMA/Ionomycin cocktail and plated on a round-bottom 96-well plate (Thermofisher Scientific). Re-suspended cells were incubated at 37 °C for 4 h. Stimulated cells were centrifuged at 1800 rpm for 4 min at 4 °C. Cell pellets were washed by topping up with 200 μl of FACS buffer and centrifugation at 1800 rpm for 4 min at 4 °C. Washed cells were stained as follows: live/dead staining for 20 min on ice; surface staining overnight at 4 °C; and intracellular staining overnight at 4 °C.

## Histology and microscopy
Loose fatty tissues were removed from shaved dorsal skin with a pair of forceps. The skin was fixed in 10% neutral buffered formalin (Fisher Scientific) overnight at 4 °C. On the following day, fixed tissues were washed twice in PBS for 5 mins each on a rocking platform. Following wash, skin tissues were either stored in 70% EtOH at 4 °C to later be embedded in paraffin. Fontana & Masson staining was performed as per manufacturer's protocol (Abcam). F&M slides were imaged using a Nanozoomer (Hamamatsu photonics) with a ×40 objective.

## Immunofluorescence and tissue microarray
Immunofluorescence labelling was performed using tissue microarrays generated using a manual tissue arrayer (Beechers Instruments). Individual tissue cores (4 mm diameter) were extracted from FFPE skin tissue blocks and moved to a recipient block using a Beechers MTA1 manual tissue arrayer (Beechers Instruments). Multiplex staining was performed using the following antibodies anti-FoxP3 (14-5773-82, Thermofisher), anti-DCT (ab74073, Abcam), and anti-CD117 (AF1356, RnD systems). Immunohistochemical experiments were conducted using Bond Max automated staining system (Leica, UK). Each antibody was optimized for pH and concentration dependence, antigen retrieval and temperature parameters. RNAscope was performed as per manufacturer's protocol. Confocal microscopy was performed with a Leica SP8 confocal microscope using a ×20 objective.

## In situ transcript quantification
RNA-scope slides were analysed using QuPath[50]. Cells were detected using the DAPI channel. Transcript counts per cell was quantified using built-in subcellular spot detection function.

## Stage-matched melanin quantification

Pigmentation index was calculated from F&M images as follows. Images were converted to 16-bit using Fiji. Otsu method thresholding was performed to filter for black pixels corresponding to melanin. Hair follicle staging was performed according to prior guidelines (REF Mueller Rover). Anagen I-II were classified as early anagen. Anagen IIIa-b were classified as mid anagen. Anagen IV-VI were classified as late anagen. Area of melanin per mid anagen hair follicle was quantified and analysed.

## Melan A cell culture with IGF-1

Melan A cells were cultured in Melan-a expansion medium (RPMI-1640 supplemented with 10% FBS and 200 nM phorbol-12-myristate-13-acetate) as per manufacturer's protocol. For IGF-1 stimulation, Melan A cells were cultured in RPMI-1640 supplemented with 10% FBS and 100 ng/ml of IGF-1 (Peprotech) for 24 h at 37 °C.

## Whole skin RNA-seq

4 mm biopsy of skin samples were stored in RNAlater (Sigma) at −20 °C. Skin samples were homogenised with Kinematica™ Polytron™ handheld PT1200E homogeniser (Fisher Scientific) using a 1200E probe (Fisher Scientific). The RNA was extracted using the mirVana™ miRNA Isolation kit (Invitrogen) by following the manufacturer's protocol. Prior to start of digestion, and between samples, the probe was sequentially washed in 1% SDS, 100% ethanol and dH2O to avoid cross-contamination. Samples were maintained on ice at all times. Extracted RNA was stored at −20 °C. Concentration of the RNA was determined using Qubit™ Broad Range assay kit (Invitrogen). Quality of the RNA was checked using Bioanalyzer (Agilent Technologies) at the King's Genomics Centre. Only samples with RNA integrity score greater than 7.0 were taken further for downstream applications. 2 μg of total RNA per samples were used for RNA-seq. 150 bp paired-end Sequencing was performed on the NovaSeq platform (Illumina) at Genewiz. Sequenced reads were trimmed using Trimmomatic v.0.36. Trimmed reads were mapped to the Mus musculus GRCm38 reference genome using the STAR aligner v.2.5.2b to produce BAM files. Unique gene counts were calculated using featureCounts function from the Subread package v.1.5.2. Only reads that fell within exon regions were counted to generate the gene hit counts table. Differential expression analysis was performed using DESeq2. Wald test was used to generate p-values and log2 fold changes.

## Pathway analysis

An enrichment analysis was performed to extract biological information from the list of pigmentation-associated genes. The enrichment analysis was performed using a web-based platform, ShinyGO (136). A false discovery rate cut-off of 0.3 was used for the list of pigmentation-associated genes. All other pathway analysis was performed with a false discovery rate cut-off of 0.05.

## Mouse skin scRNA-seq

Skin CD45+ and CD45- cells were FACS-sorted into RPMI-1640 supplemented with 40% heat-inactivated FBS and 40 U/ml of RiboLock RNase inhibitor (Thermofisher). The cells numbers were counted and mixed at 2:1 ratio (CD45+:CD45-). The mixed cells were used for cDNA library generation using the 10X Genomics Chromium Single-cell 3′ kit by the Genomics Centre at the Blizard Institute. Prepared libraries were sequenced using the NovaSeq 6000 S4. Raw sequencing data were processed using the 10X Genomics *CellRanger* package.

## Treg bulk RNA-seq

Skin tissues and draining lymph nodes were harvested from 9 days old mice and digested as described above. Following digestion, 3 biological replicates were pooled into 1 sample (3 samples in total). Pre-selection was performed by MACS using CD45 microbeads (Miltenyi) as per manufcaturer's protocol. Finally, DAPI- CD45+ CD4+ Foxp3-EGFP+ Tregs were FACS-sorted from the skin and the lymph nodes. Cells were directly sorted into TRIzol™ LS (Thermofisher) and stored at −80 °C. Only samples with RNA integrity score greater than 7.0 were taken further for downstream applications. 2 μg of total RNA per samples were used for RNA-seq. 150 bp paired-end Sequencing was performed on the NovaSeq platform (Illumina) at Genewiz. Sequenced reads were trimmed using Trimmomatic v.0.36. Trimmed reads were mapped to the Mus musculus GRCm38 reference genome using the STAR aligner v.2.5.2b to produce BAM files. Unique gene counts were calculated using featureCounts function from the Subread package v.1.5.2. Only reads that fell within exon regions were counted to generate the gene hit counts table.

## Human skin scRNA-seq

Human neonatal and adult melanocyte scRNA-seq data is available in the Gene Expression Omnibus (GEO) database repository under accession number GSE151091. Human healthy and vitiligo skin scRNA-seq data is available in the Genome Sequence Archive (GSA) with accession number PRJCA006797. Analysis was performed using Seurat package in the R programming environment.

## Statistical analysis

Statistical analysis was performed using GraphPad Prism 8.0 (GraphPad Inc.). Unpaired t-test, paired t-test or one way ANOVA was performed as indicated in figure legends to calculate p values. Statistical significance was inferred if p value was less than 0.05 (*), 0.01 (**), 0.001 (***), or 0.0001 (****). P values greater than 0.05 was identified as not statistically significant.

## Reporting summary

Further information on research design is available in the Nature Portfolio Reporting Summary linked to this article.

## Data availability

Bulk RNA and scRNA-seq data are available in the GEO database repository with accession numbers GSE309398, GSE309399, and GSE309400. Human neonatal and adult melanocyte scRNA-seq data is available in the GEO database repository under accession number GSE151091. Human healthy and vitiligo skin scRNA-seq data is available in the GSA with accession number PRJCA006797. Source data are provided with this paper.

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

## Acknowledgements

We gratefully acknowledge the Advanced Cytometry Platform (Flow Core), Research and Development Department at Guy's and St Thomas' NHS Foundation Trust; the imaging facility staff at University College London; the Flow Cytometry Facility at Barts Cancer Institute; and the Genomics Centre at the Blizard Institute, Queen Mary University of London. We also thank Aamir Ahmed and Mike Millar (TissuePlexia) for their assistance with tissue microarrays, Julian Downward for generously providing the Melan A cell line, and Abigail Tucker for kindly providing the K14-Cre mice. This work was supported by Wellcome Trust grants 213401/Z/18/Z to N.A., 220009/Z/19/Z to I.C., and 224910/Z/21/Z to J.Z.X.

## Author contributions

Conceptualization: N.A. Methodology: I.C., N.A., H.A., B.X., H.J., and J.C. Investigation: I.C., H.A., J.X., P.L., M.S., P.T., J.T.W., and S.A. Visualization: I.C., and H.A. Funding acquisition: N.A. Project administration: N.A.

Supervision: N.A. Writing – original draft: N.A. and I.C. Writing – review & editing: N.A. and I.C.

## Competing interests

The authors declare no competing interests.
