## [Transparent Peer Review file · Nature Communications]

Early skin seeding regulatory T cells modulate PPAR γ -dependent skin pigmentation

Corresponding Author: Dr Niwa Ali

Version 0:

Reviewer comments:

Reviewer #1

(Remarks to the Author)

In this manuscript, Cho and colleagues showed that Treg cells seeding the skin during the first two weeks of life are instrumental to the normal function of melanocyte stem cells (MeSCs) and skin pigmentation in early life. Mechanistically, the presence of skin Tregs is required to activate the PPAR γ signal pathway in hair follicle (HF) cells, which in turn promotes MeSC to produce melanin. In human vitiligo patients, HF cells at the skin lesion also showed downregulation of PPAR-g activity. Overall, this is an exciting finding linking neonatal skin Tregs to MeSC function and skin pigmentation. However, the overall enthusiasm for the manuscript was dampened by major gaps in the current study and other significant technical issues.

Major points:

1. There are two major gaps in the current study. First, it is not clear how ETregs activate the PPAR γ pathway in HF cells. Is it direct or indirect? If it is direct, what is the factor(s) produced by ETregs that activates PPAR γ in HF cells?
2. Second, how does activation of the PPAR γ pathway in HF cells instruct MeSCs to upregulate melanin production-related genes? Again, which factor(s) produced by HF cells promote MeSC to produce melanin?
3. Although DT was only injected on P6 and P8, Treg numbers will take a longer time to recover. Please show Treg percentage and number in more time points between P9 and P28. Similarly, skin CD4 and CD8 T cells' numbers and cytokine profiles should also be analyzed at more time points between P9 and P28 (Fig. 3F-3M).
4. In sFig 3D and 3E, it is not clear how MeSC proliferation rates in control mice were much higher compared to dTreg mice, yet the numbers of MeSCs were similar between the two groups.
5. Please perform RT-qPCR to detect PPAR γ target genes from sorted HF cells to validate the imaging results in Fig. 3I-3L.
6. In both the PPAR γ antagonist and agonist experiments in Fig. 4, the drugs targeted every tissue in the body. These treatments need to be applied to PPAR γ HF cell-specific KO mice to validate the conclusions.
7. The human skin scRNA data do not support the conclusions from the neonatal mouse model. Vitiligo is an autoimmune disorder. The authors didn't show that Treg numbers in the patient skin samples were reduced similar to those in the DT-treated neonatal mouse (Fig. 6A). Fig. 6C-6E showed human MeSCs express a significant number of PPAR γ target genes, but in the mouse scRNA data, no differences were observed in PPAR γ target gene expression in the MeSCs between control and dTreg mice (Fig. 5E, 5I).

Minor points:

1. Figure legend for Fig. 2E is not correct.
2. Please provide differential expressed gene lists for Figure 3B, 3C, and 3G.
3. The references to Fig. 6A to 6E were in the wrong order (Lines 479, 482, 495, and 500)

Reviewer #2

(Remarks to the Author)

In the manuscript by Cho et.al., the authors investigated the impact of depleting regulatory T cells on the pigmentation of mice's back skin. Using a DTR system controlled by the FoxP3 promoter, Foxp3⁺ Tregs were depleted during specific postnatal periods, leading to notable changes in skin pigmentation in P28 mice. While the phenotype is interesting, a more comprehensive analysis is necessary to discern the primary driver of these changes. The current explanation of the phenotype cannot rule out the influence of the hair cycle.

Major issues:

1. In adult mice, the back skin lacks melanocyte lineage cells, which are exclusively housed in the hair follicles. Therefore, the observed phenotype pertains to pigmentation changes within the hair follicles rather than on the skin surface. In the hair follicles, this pigmentation variation aligns with the hair cycle (telogen-anagen-catagen), with mature melanocytes secreting pigments only at the anagen phase. In mice where ETreg depletion occurs, it is clear that the follicles remain in the dormant telogen state, contrasting with control mice in the active anagen phase, which could result in distinct skin color. Thus, it seems that ETreg depletion predominantly impacts the hair cycle rather than directly affecting melanocyte lineages. To solve this problem, a more detailed analysis of the changes in the hair cycle and the functionality of melanocyte lineage cells with accurate immunostaining is needed.
2. The RNAseq analysis of whole skin samples cannot capture specific changes in the melanocyte lineages. During the anagen phase, the entire skin undergoes significant reorganization, altering the transcriptome of various cell types, such as adipocytes. Consequently, the identified PPAR pathways could be attributable to this global reorganization rather than changes in the melanocyte lineages.
3. To enhance the robustness of the study, improvements are needed in the quality of immunostaining images. Presently, it is challenging to discern the indicated cell types, particularly the melanocyte lineages labeled by DCT. Furthermore, it is crucial to utilize immunostaining to accurately determine changes in Treg numbers.
4. The quantification method for melanocyte lineage cells presents issues as it also identifies CD117⁺ hair bulb cells within the hair follicles.
5. The administration of PPAR γ agonist and antagonist appears to be more linked to the hair cycle rather than to the melanocyte lineage specifically.
6. The intraperitoneal injection of DT makes it challenging to discern between the systemic and local effects of Treg depletion.

Reviewer #3

(Remarks to the Author)

The study provides an intriguing exploration into the immunological mechanisms involved in skin pigmentation, particularly focusing on the role of regulatory T cells (Tregs) and the peroxisome proliferator-activated receptor- γ (PPAR γ) pathway. The study investigates the temporal dynamics of Treg involvement in the skin's immune environment during early neonatal development and its implications for melanocyte stem cell (MeSC) mediated pigmentation.

The research employs an array of techniques such as flow cytometry, transcriptomic analysis, single-cell RNA-sequencing, and in vivo experiments. The findings potentially unravel new pathways for understanding skin pigmentation disorders and highlight a critical period in early development where Tregs play a significant role in establishing skin homeostasis and pigmentation patterns. The link established between ETregs and PPAR γ activity in MeSCs during postnatal development and the paralleled disruptions observed in the human skin disorder vitiligo provide a strong translational significance to the study.

However, for a more in-depth assessment, the following questions need to be addressed.

1. How are the specific activation markers identified to assess Tregs and their relevance to the study's context?
2. In the manuscript, Tregs are described as accumulating in the skin by postnatal day 13. What mechanisms are proposed for this accumulation, and is there evidence to suggest a selective recruitment or retention of Tregs in the skin?
3. The study mentions the influx of Tregs in skin during the early neonatal period; could this be due to local proliferation or migration from peripheral sites?
4. How does the P3-P12 interval influence the functional status of Tregs in the skin?
5. Is there a possibility that the punctual depletion of ETregs at P6-P8 has immediate versus delayed effects on MeSCs?
6. Could the study's findings about PPAR γ signaling activity be applicable to other stem cell niches in the skin, such as the HFSCs?
7. The manuscript suggests that melanocyte function is regulated by Tregs independent of secondary mediators. Can authors provide more details on this regulatory mechanism?
8. What are the implications of the transient loss of Tregs on systemic inflammation and how does it correlate with skin

pigmentation?

9. Can authors further explain the importance of a specific 'window' of time for Treg requirement in skin pigmentation?
10. How does the transcriptomic dysregulation of the PPAR γ signaling pathway immediately post ETreg depletion manifest on a cellular level?
11. What additional immune cell subsets could be influenced by ETreg depletion that were not assessed in this study?
12. Is there potential redundancy in the signaling pathways that regulate MeSC function, and could other pathways compensate for the loss of PPAR γ signaling?
13. How does the involvement of Tregs in the suppression of type 2 helper T cell mediated fibrous pathology contribute to skin pigmentation?
14. What is the functional role of Tregs in supporting HFSC differentiation, and how does this interact with MeSC-mediated skin pigmentation?
15. Can the results observed in the mouse model regarding ETreg depletion and the PPAR γ pathway be translated to human skin development or skin pigmentation disorders directly?
16. The manuscript posits that the PPAR γ pathway is critical for melanocyte function. Are there other pathways with potential involvement in this process that warrant further investigation?
17. Given that ETregs were found to be essential for HF PPAR γ signaling, what are the broader implications for the understanding of immune cell involvement in tissue development?
18. How would the findings of this study influence therapeutic approaches to skin pigmentation disorders such as vitiligo?
19. The manuscript reports a reduction in MeSC proliferation upon ETreg depletion. Could this be linked to specific changes in the cell cycle or apoptotic pathways?
20. How does the heterogeneity of Tregs, possibly containing subsets with distinct roles and activation states, affect the overall interpretation of the results?
21. Can authors discuss the significance of the timing of ETreg depletion on P6-P8 and how this differs from potential interventions at other developmental stages?
22. The authors discussed the role of ETregs in melanocyte stem cell function. Can authors provide more details on the experimental evidence supporting a direct interaction between these cells?
23. How does the study address the potential compensatory mechanisms that may arise from the depletion of ETregs?
24. How does the study address the heterogeneity within Treg and MeSC populations, and could there be subpopulations with distinct roles?
25. In Figures 3B and 2C, the color legends “-log₁₀FDR” should be corrected as “-log₁₀(FDR)”.
26. It is unclear how the differentially expressed genes (DEG) were identified. Which test/statistical model was used? Is it comparing one cell type and the pool of all the remaining cell types, or is it comparing any pair of cell types and then aggregate in some way? Can authors explain why some cell types do not show any DEGs?
27. In Figures 5K and 6E, the gene expression levels range from -3000 to more than 2000? Did authors normalize the data by sequencing depth and perform any reasonable transformation? This will affect the robustness of the single-cell RNA-seq analysis.
28. Authors stated that “Neonatal Tregs regulate the hair follicle transcriptome and PPAR γ activity”. What evidence supports the causality between neonatal Treg presence and changes in the hair follicle transcriptome, rather than a correlation or secondary consequence of other developmental processes?
29. How do authors rule out the possibility that observed transcriptomic changes in the hair follicle are independent of Treg activity and could be attributed to the intrinsic maturation of the hair follicle itself?
30. Can the study distinguish between direct effects of Tregs on hair follicles versus indirect effects mediated by other cell types or signaling molecules in the skin microenvironment?
31. What methodologies were used to exclude the influence of global developmental changes occurring in neonatal skin that might confound the specific attribution to Tregs?
32. How were alternative pathways or transcription factors that might regulate the PPAR γ pathway in hair follicles independently of Tregs accounted for in the analysis?
33. Were the findings validated in vitro, such as in hair follicle organ cultures, where the Treg influence can be modulated independently of the in vivo environment?
34. In the single-cell analysis, how was the specificity of PPAR γ activity scores to Treg interactions ascertained against the background of the complex skin cellular milieu?
35. Are there data to show that manipulation of Treg levels or function does not alter the developmental timing of hair follicle maturation, which could indirectly affect transcriptomic profiles?
36. What are the temporal aspects of the Treg impact on hair follicle transcriptomics? Is this a transient phase during neonatal development, or are there long-lasting effects?

Version 1:

Reviewer comments:

Reviewer #1

(Remarks to the Author)

In this revision, the authors provided a large amount of new data, including new RNA-seq experiments showing the

molecular links between ETreg, HFSC, and MeSC, the keratinocyte-specific PPAR γ conditional knockout mouse showing PPAR γ 's unique role in HFSCs. The text of the manuscript has been revised extensively to be more accurate and clearer, addressing the reviewer's previous concerns. These changes have significantly elevated the quality of the manuscript. The reviewer shares the excitement of the authors' surprising findings that established a novel link between neonatal skin Treg cells and MeSC-mediated skin pigmentation.

Reviewer #2

(Remarks to the Author)

Thank you for your detailed responses to the review comments. Overall, your revisions effectively address the points raised. The key supplemental experiments resolve the primary concern regarding potential hair cycle confounding of the pigmentation phenotype, and the new sequencing data provide additional support for the role of keratinocyte-intrinsic PPAR γ signaling in regulating melanocyte activity.

I acknowledge the technical challenges encountered with DCT staining and support the removal of suboptimal immunofluorescence images to enhance data clarity. For future melanocyte visualization studies, SOX10 or S100 could be considered as alternative markers. Note that these also label Schwann cells, but distinction may be achievable through location-specific assessment combined with morphological evaluation.

The manuscript shows clear improvement in its revised form.

Reviewer #3

(Remarks to the Author)

The authors' time and efforts to address the previous comments are appreciated. Most of my concerns have been satisfactorily addressed. However, the following points remain:

For Comment 11: Can authors incorporate the discussion on limitations into the Discussion section?

For Comment 26: Can authors provide full details of the methods used in the Methods section to ensure reproducibility?

For Comment 27: For Figures 5K and 6E, please clearly indicate that the values are scaled for visualization purposes only. Also, describe the scaling or transformation procedures used; otherwise, the interpretation of the values may be misleading.

For Comment 29: Please consider including the results of this analysis in the Supplementary Materials.

Re: Manuscript # NCOMMS-24-09213,
" Early skin seeding regulatory T cells modulate PPAR γ -dependent skin pigmentation"

Reviewer #1:

We appreciate the comments on our manuscript from Reviewer 1, who states:

"Cho and colleagues showed that Treg cells seeding the skin during the first two weeks of life are instrumental to the normal function of melanocyte stem cells"

"Overall, this is an exciting finding linking neonatal skin Tregs to MeSC function and skin pigmentation."

Major Points

Comment 1:

"There are two major gaps in the current study. First, it is not clear how ETregs activate the PPAR γ pathway in HF cells. Is it direct or indirect? If it is direct, what is the factor(s) produced by ETregs that activates PPAR γ in HF cells?"

We thank the reviewer for highlighting this important question regarding the mechanistic link between early Tregs (ETregs) and PPAR γ activation in hair follicle (HF) cells. We agree that the direct functional connection remained incompletely addressed in our initial submission. Due to the very low abundance of ETregs in the skin during P6–P8 and the difficulty in maintaining their viability in conventional Treg culture conditions (e.g., IL-2-supplemented media), direct co-culture experiments with hair follicle cells were technically unfeasible. To circumvent these limitations, we implemented several complementary approaches to gain mechanistic insights into Treg–HF communication:

A) Bulk RNA-seq of Skin and SDLN Tregs

To examine whether skin-resident Tregs express candidate factors that may activate the PPAR γ pathway in HF cells, we isolated Tregs from both neonatal skin and skin-draining lymph nodes (SDLNs) of Foxp3-EGFP reporter mice at P9 for bulk RNA-seq. While Tregs were underrepresented in our whole skin single cell RNA-seq data, bulk RNA-seq enabled robust profiling of these rare populations. The complete dataset will, of course, be made publicly available.

B) NicheNet Analysis to Predict Ligand–Receptor Interactions

To determine how skin Tregs may influence PPAR γ activity in HF keratinocytes, we next performed NicheNet analysis to predict ligand–receptor interactions between skin Tregs (sender cells) and HF keratinocytes (receiver cells). This revealed Il10 and Tnfsf9 as top candidate Treg-derived ligands, predicted to induce expression of Apoe, Abca1, and Pltp in HF keratinocytes - genes known to regulate lipid uptake and activate the PPAR γ pathway (**New Figures 5G, 5K**). Previous work (<https://doi.org/10.1016/j.neuron.2014.11.020>) supports the role of IL-10 and TNFSF9 in regulating these targets. Furthermore, our bulk RNA-seq data show that Il10 and Tnfsf9 are preferentially expressed in skin Tregs compared to SDLN Tregs (**New Figure 5L**), suggesting a tissue-specific mechanism by which skin Tregs may promote PPAR γ activity in HF cells.

C) Downstream Mediators: Linking Keratinocytes to Melanocytes via IGF-1:

Having established a potential Treg–keratinocyte axis, we next asked how PPAR γ activation in keratinocytes may influence melanocyte stem cell (MeSC) function. Using CellChat analysis of our scRNA-seq data, we identified a ligand–receptor interaction between HF keratinocytes and melanocytes via IGF-1 (insulin growth factor-1), a factor known to promote melanin synthesis (**New Figure 5G-H**; see also: <https://doi.org/10.1007/s11626-016-0052-y>).

To functionally test this, we cultured Melan-A cells in the presence of IGF-1 and observed increased melanin production (**New Figure 5I–J**). This result suggests that IGF-1 produced by keratinocytes downstream of PPAR γ signaling can directly enhance melanocyte pigmentation. In support of this model, IGF-1 is a known transcriptional target of PPAR γ (see: <https://doi.org/10.1371/journal.pone.0173174>), and our scRNA-seq data show that IGF-1 expression is reduced in HF keratinocytes following ETreg depletion. Together, these data support a model in which skin Tregs, via IL-10 and TNFSF9, promote PPAR γ signaling in HF keratinocytes, which in turn drives IGF-1–mediated stimulation of MeSC pigmentation.

Comment 2:

“Second, how does activation of the PPAR γ pathway in HF cells instruct MeSCs to upregulate melanin production-related genes? Again, which factor(s) produced by HF cells promote MeSC to produce melanin?”

We thank the reviewer for this insightful follow-up question. As discussed in our response to Comment 1, we propose that PPAR γ activation in hair follicle (HF) keratinocytes leads to increased expression of IGF-1, a well-established transcriptional target of PPAR γ (doi.org/10.1371/journal.pone.0173174). In our scRNA-seq dataset, IGF-1 expression is downregulated in the HF compartment upon ETreg depletion, consistent with reduced PPAR γ activity. To functionally test the role of IGF-1 in melanocyte activation, we cultured a mouse melanocyte cell line (Melan-A) in the presence of recombinant IGF-1. This resulted in a significant increase in pigmentation (**New Figure 5I–J**), consistent with previous findings demonstrating that IGF-1 promotes melanin synthesis (<https://doi.org/10.1007/s11626-016-0052-y>). Taken together, these data support a model in which PPAR γ activation in keratinocytes induces IGF-1 expression, which in turn acts on melanocyte stem cells (MeSCs) to promote melanin production.

Comment 3:

“Although DT was only injected on P6 and P8, Treg numbers will take a longer time to recover. Please show Treg percentage and number in more time points between P9 and P28. Similarly, skin CD4 and CD8 T cells’ numbers and cytokine profiles should also be analyzed at more time points between P9 and P28 (Fig. 3F–3M).”

We appreciate the reviewer’s suggestion to examine the recovery period between P9 and P28 following diphtheria toxin treatment on P6 and P8. Although our focus was on the immediate impact of early Treg depletion (with transcriptomic changes already evident by P9 in bulk RNA-seq and RNAscope analysis), we agree that additional profiling during the recovery phase may provide useful context.

To address this, we performed flow cytometric analysis of cytokine-producing T cells isolated from both the skin and skin-draining lymph nodes (SDLNs) on postnatal day 17 (P17), following Treg depletion on P6 and P8.

Our results show that in the SDLN, there is an increase in IFN gamma positive CD8 T cells and IL-13 positive TCR gamma delta T cells, alongside a reduction in IL-17 positive TCR gamma delta T cells and TNF alpha positive FoxP3 negative CD4 effector T cells. In the skin, we observed elevated expression of IFN gamma, IL-13, and TNF alpha in dermal gamma delta T cells and effector CD4 T cells, along with reduced IL-2 production in dendritic epidermal T cells. These results are presented in New Supplementary Figure 7 and collectively demonstrate a sustained inflammatory response both locally and systemically at P17 following early Treg loss.

However, based on our finding that modulation of PPAR gamma signaling at early time points is sufficient to alter skin pigmentation, and that co-depletion of CD8 T cells does not affect this phenotype, we believe that events occurring during the P9 to P28 period are unlikely to be the primary drivers of pigmentation changes. Therefore, while these data further support the inflammatory consequences of early Treg depletion, they do not alter the central conclusions of our manuscript.

Comment 4:

“In sFig 3D and 3E, it is not clear how MeSC proliferation rates in control mice were much higher compared to dTreg mice, yet the numbers of MeSCs were similar between the two groups”

We thank the reviewer for this valuable observation. We now provide further explanation of this point in the main text to clarify our interpretation for the reader (see page 5, paragraph 2 of revised manuscript). As previously reported (<https://www.nature.com/articles/s41586-023-05960-6>), melanocytes undergo turnover during the anagen phase of the hair cycle. We therefore hypothesise that although MeSCs in control mice exhibit higher proliferation rates, this may be balanced by concurrent turnover of differentiated melanocytes, resulting in a steady-state population size. This could explain why melanocyte numbers appear similar between control and dTreg mice despite differences in proliferation.

In addition, we acknowledge a technical limitation in our flow cytometry approach. Certain melanocyte subsets, particularly those residing in the hair bulb, may be lost during sample preparation, and thus not captured in our analysis. We have now clarified this caveat in the manuscript and have moderated our claims regarding changes in melanocyte number and proliferation accordingly (see page 5, paragraph 2 of revised manuscript).

Comment 5:

“Please perform RT-qPCR to detect PPAR γ target genes from sorted HF cells to validate the imaging results in Fig. 3I-3L”

We thank the reviewer for this helpful suggestion. Unfortunately, we do not currently have access to a reporter mouse line that would enable us to reliably isolate hair follicle (HF) epithelial cells for RT-qPCR analysis. As a result, direct sorting of HF cells and subsequent validation via RT-qPCR is technically challenging in our system. Nonetheless, we fully agree that it is important to assess the expression of PPAR γ target genes using complementary methods beyond RNAscope. To this end, we leveraged our whole skin single-cell RNA-seq dataset to examine PPAR γ target gene expression in the HF compartment. We first curated a list of functionally validated PPAR γ target genes from a previously published database (doi.org/10.1155/2016/6042162). We then identified differentially expressed genes in hair follicle epithelial cells between Treg-sufficient and Treg-depleted conditions and subset this list based on known PPAR γ targets. This analysis revealed that multiple PPAR γ target genes were downregulated in the HF compartment following Treg depletion (**New Figure 5G**)

We note that some key targets identified by RNAscope, such as **Fabp7 and Adipoq**, were not recovered as differentially expressed in the scRNA-seq data. This discrepancy may be due to several technical or biological factors, including:

- Low expression levels that fall below the detection threshold in single-cell RNA-seq
- Spatial restriction of expression to specific HF layers or niches that are underrepresented in dissociated single-cell preparations
- Dropout effects or loss of transcripts during tissue dissociation and library preparation
- Batch-related variation between scRNA-seq and RNAscope sample sets
- The possibility that expression is regulated post-transcriptionally or is more dynamic than captured at the single time point analyzed

Despite these limitations, the overall direction of change across multiple datasets and transcriptional downregulation of validated PPAR γ targets in HF epithelial cells, supports the conclusion that PPAR γ signaling is reduced following Treg depletion.

To further substantiate this, we refer the reviewer to new experiments described in Comment 6 (below), in which we generated and analyzed K14Cre PPAR γ flox mice. These animals exhibit a clear loss of pigmentation, providing direct genetic evidence that PPAR γ signaling in hair follicle keratinocytes is essential for normal skin pigmentation. This strongly supports the central mechanism proposed in our study.

Comment 6:

“In both the PPAR γ antagonist and agonist experiments in Fig. 4, the drugs targeted every tissue in the body. These treatments need to be applied to PPAR γ HF cell-specific KO mice to validate the conclusions”

We thank the reviewer for this important point. To directly address the concern about systemic drug effects and to determine whether pigmentation is dependent on PPAR γ activity within the hair follicle (HF) epithelium, we generated a new, targeted genetic model. Specifically, we crossed Krt14-Cre mice with Pparg^{flox/flox} mice to produce Krt14-Cre;Pparg^{flox/flox} (hereafter referred to as K14-Pparg^{KO}) animals. This model enables constitutive deletion of Pparg in basal keratinocytes, including those of the HF epithelium, from early development onwards. The generation and use of this multi-allelic transgenic line represent a substantial technical addition to the study and allow us to isolate the role of epithelial-intrinsic PPAR γ in regulating pigmentation, independent of systemic pharmacologic effects.

To assess pigmentation in this model, we performed Fontana-Masson staining on dorsal skin sections collected at P28 from both K14-Pparg^{KO} and littermate K14-Pparg^{WT} control mice. As shown in **New Figure 4H-I**, we observed a marked reduction in melanin content in K14-Pparg^{KO} skin relative to controls. Quantification of the melanin-positive area revealed a significant decrease in pigmentation in the knockout mice, confirming that epithelial PPAR γ signaling is required for normal skin pigmentation.

This experiment provides definitive genetic evidence that the effects observed with systemic PPAR γ agonist and antagonist treatments are not off-target or indirect, but instead reflect a *bona fide* requirement for PPAR γ within the HF epithelial compartment. These findings strengthen the core conclusion of our study by demonstrating that PPAR γ acts cell-intrinsically in the skin epithelium to regulate melanin production.

For space and aesthetic consistency, we have removed the original images of mouse skin while retaining the Fontana-Masson histology and quantification in the main figure.

Comment 7:

“The human skin scRNA data do not support the conclusions from the neonatal mouse model. Vitiligo is an autoimmune disorder. The authors didn’t show that Treg numbers in the patient skin samples were reduced similar to those in the DT-treated neonatal mouse (Fig. 6A). Fig. 6C-6E showed human MeSCs express a significant number of PPAR γ target genes, but in the mouse scRNA data, no differences were observed in PPAR γ target gene expression in the MeSCs between control and dTreg mice (Fig. 5E, 5I).”

We appreciate this comment and the opportunity to clarify the relationship between our mouse and human datasets. We acknowledge that the number of melanocytes captured in our mouse scRNA-seq data is low, limiting statistical power to detect differentially expressed genes in this population. Nevertheless, we performed differential expression analysis between melanocytes from control and ETreg-depleted mice. After correction for multiple testing, no genes reached statistical significance (adjusted p-values > 0.05), likely due to the small cell numbers.

However, when examining unadjusted p-values, melanocytes showed the highest number of differentially expressed genes across all cell types following ETreg depletion, suggesting that a true biological signal may be present but underpowered for detection. For transparency, we have included a list of the top 200 differentially expressed genes for reviewer reference (appended to the end of this letter), but have not incorporated these results into the revised manuscript due to their exploratory nature.

Regarding the human vitiligo dataset: these data were not generated in our lab, and we do not have access to additional samples or metadata (such as Treg quantification). The dataset we analyzed is a publicly available single-cell RNA-seq dataset from vitiligo lesional and non-lesional skin. Our intention was not to suggest that Treg numbers are necessarily reduced in these patients, but rather to explore whether melanocytes in vitiligo lesional skin show transcriptional patterns suggestive of impaired PPAR γ activity, similar to what we observed in our neonatal Treg-depleted mouse model. We now clarify in the manuscript that this human analysis is intended as exploratory and hypothesis-generating, not as direct mechanistic validation (see page 17, paragraph 1 of revised manuscript). Specifically, we propose that early-life dysregulation of Treg function may impair melanocyte homeostasis in a way that is relevant to human pigmentation disorders such as vitiligo, but we do not claim that the pathogenic mechanisms are identical.

Minor points:

1. Figure legend for Fig. 2E is not correct.

This has now been corrected.

2. Please provide differential expressed gene lists for Figure 3B, 3C, and 3G. This is now provided as a supplementary table.

3. The references to Fig. 6A to 6E were in the wrong order (Lines 479, 482,495, and 500) Many thanks for careful reading of our manuscript, this has also been corrected.

Reviewer #2:

Major Points

Comment 1:

“In adult mice, the back skin lacks melanocyte lineage cells, which are exclusively housed in the hair follicles. Therefore, the observed phenotype pertains to pigmentation changes within the hair follicles rather than on the skin surface. In the hair follicles, this pigmentation variation aligns with the hair cycle (telogen-anagen-catagen), with mature melanocytes secreting pigments only at the anagen phase. In mice where ETreg depletion occurs, it is clear that the follicles remain in the dormant telogen state, contrasting with control mice in the active anagen phase, which could result in distinct skin color. Thus, it seems that ETreg depletion predominantly impacts the hair cycle rather than directly affecting melanocyte lineages. To solve this problem, a more detailed analysis of the changes in the hair cycle and the functionality of melanocyte lineage cells with accurate immunostaining is needed.”

To specifically address the concern that ETreg depletion may delay anagen onset, and that this might explain the pigmentation phenotype, we performed the following additional experiments:

1. We depleted Tregs on P6 and P8, as in our original model, and harvested skin at three later time points: P28, P29, and P30. This allowed sufficient time for hair follicles in the Δ ETreg condition to enter late anagen, thus matching the stage observed in control mice on P28.
2. We then performed Fontana-Masson staining on these samples and quantified melanin abundance specifically within anagen-matched hair follicles, across conditions. Despite matching for hair cycle stage, we observed a significant reduction in melanin content in Δ ETreg follicles, as shown in the newly included quantification (**New Figure 2D-E**).

This experiment directly addresses the reviewer’s concern. The persistence of the pigmentation defect even after allowing for delayed anagen entry suggests that the phenotype is not due to a shift in hair cycle timing, but instead reflects a true functional defect in melanocyte lineage cells following early-life Treg depletion.

We also acknowledge the reviewer’s suggestion to perform more detailed immunostaining to assess melanocyte lineage functionality. Markers such as DCT would indeed be ideal; however, despite extensive efforts across two laboratories, including a dedicated histology core facility, we were unable to obtain reliable DCT staining in neonatal and early postnatal mouse skin. Optimisation was carried out using both OCT and paraffin-embedded sections, a range of antigen retrieval methods, and two different commercially available DCT antibodies.

Unfortunately, the most widely used antibody in the field (goat anti-DCT, Santa Cruz sc-10451) has been discontinued and is no longer accessible. This antibody has been employed in high-profile studies, including those from laboratories specialising in melanocyte biology (e.g., <https://www.nature.com/articles/s41586-023-05960-6>). Newer antibodies tested did not produce consistent staining in our hands, even after optimisation.

Given these limitations, we instead focused on a physiologically relevant and robust readout: quantitative analysis of melanin content in hair cycle–matched follicles. We believe this approach offers a reliable surrogate for melanocyte function *in vivo* and enables us to compare pigmentation outcomes independently of follicle stage.

Comment 2:

“The RNAseq analysis of whole skin samples cannot capture specific changes in the melanocyte lineages. During the anagen phase, the entire skin undergoes significant reorganization, altering the transcriptome of various cell types, such as adipocytes. Consequently, the identified PPAR pathways could be attributable to this global reorganization rather than changes in the melanocyte lineages”

We thank the reviewer for raising this important point. We fully agree that bulk RNA sequencing of whole skin cannot resolve cell type specific transcriptomic changes, and that global transcriptional changes during the anagen phase could mask or confound signals from individual lineages such as melanocytes. In particular, PPAR gamma target genes are expressed by multiple skin cell types, including adipocytes. Therefore, observed changes in these transcripts in whole skin cannot be specifically attributed to melanocytes. To address this limitation, we performed RNAscope for the PPAR gamma target genes *Fabp7* and *Adipoq*, and observed decreased expression in the hair follicle compartment following Treg depletion. However, we acknowledge that this approach still does not allow for definitive assignment to specific cell types. We have now made this limitation more explicit in the revised manuscript to provide appropriate context for these results. (see page 9, paragraph 3 of revised manuscript).

That said, certain transcripts such as *Dct* and *Oca2* are, to our knowledge, restricted to melanocytes within the skin. We observed downregulation of both genes in whole skin bulk RNA sequencing data following early life Treg depletion. While this does not definitively localize the effect, it is consistent with the idea that melanocyte function is disrupted in this setting.

To complement the bulk and RNAscope data, we also performed single cell RNA sequencing of neonatal skin. In this dataset, we observed a general decrease in the expression of PPAR gamma target genes within the hair follicle keratinocyte population in Treg depleted mice. Furthermore, we now provide new genetic evidence that PPAR gamma activity in keratinocytes is functionally required for pigmentation. Mice lacking PPAR gamma in keratinocytes (*Krt14 Cre Pparg flox*) show significantly reduced melanin content in hair follicles (see Reviewer 1, Comment 6 above, and **New Figure 4H–I**), reinforcing our model that keratinocyte intrinsic PPAR gamma signaling plays a key role in regulating melanocyte activity.

Comment 3:

“To enhance the robustness of the study, improvements are needed in the quality of immunostaining images. Presently, it is challenging to discern the indicated cell types, particularly the melanocyte lineages labeled by DCT. Furthermore, it is crucial to utilize immunostaining to accurately determine changes in Treg numbers.”

We thank the reviewer for this helpful comment regarding the quality and interpretability of our immunostaining images. In response, we made extensive efforts to improve the visualization of melanocyte lineage cells using wholemount immunofluorescence microscopy and by testing multiple DCT antibodies across a range of conditions. Unfortunately, despite repeated optimisation attempts, we were unable to achieve consistent and high quality staining of DCT positive melanocytes in neonatal and early postnatal skin. We suspect this may be due to antigen accessibility or fixation sensitivity at these developmental stages (please also see our response to Comment 1 above)

Given these technical limitations and in the interest of maintaining the clarity and robustness of the dataset, we have removed the original Figure 3E and 3F, as well as Supplementary Figures 7A to 7C, from the revised manuscript. We believe that this decision strengthens the overall presentation and avoids overinterpretation of technically challenging data. Importantly, this change does not affect the main

conclusion of the study. Our central finding, that early life Tregs regulate PPAR gamma activity in the hair follicle, which in turn is required for melanocyte function and pigmentation, is supported by multiple independent lines of evidence, including scRNA sequencing, RNAscope, and genetic deletion of PPAR gamma in keratinocytes.

Comment 4:

“The quantification method for melanocyte lineage cells presents issues as it also identifies CD117+ hair bulb cells within the hair follicles.”

We agree with the reviewer that CD117 staining lacks the specificity to exclusively label melanocyte lineage cells and may include other cell types within the hair bulb. As noted above, we attempted alternative strategies including DCT immunostaining to more specifically visualize melanocytes, but these approaches yielded inconsistent results and were ultimately not included in the revised manuscript. Given these limitations, we have removed figures based on less specific or technically suboptimal markers. We now focus on more robust and interpretable readouts of melanocyte function, including pigmentation quantification within stage matched follicles and transcriptional analysis of melanocyte specific genes.

Comment 5:

“The administration of PPARγ agonist and antagonist appears to be more linked to the hair cycle rather than to the melanocyte lineage specifically”

We thank the reviewer for this thoughtful point. To minimise confounding by hair cycle differences, all quantification of melanin content following PPARγ modulation was performed on hair cycle stage–matched follicles (staging performed as per guidelines by Muller-Rover and colleagues; available at 10.1046/j.0022-202x.2001.01377.x). This experimental design controls for follicular timing and allows us to attribute the observed pigmentation differences to pathway modulation rather than changes in hair cycle progression. Inhibition of PPARγ signalling recapitulated the hypopigmentation seen in ΔETreg mice, while PPARγ activation rescued pigmentation despite Treg depletion (New Figure 4C), strongly suggesting a functional role for this pathway in regulating melanocyte activity. While we cannot completely exclude secondary effects on the hair cycle, these results support a direct requirement for PPARγ signalling in the pigmentation process.

Comment 6:

“The intraperitoneal injection of DT makes it challenging to discern between the systemic and local effects of Treg depletion.”

We thank the reviewer for raising this important point. To distinguish between systemic and local roles of Tregs, we administered the S1PR antagonist FTY720 on P6 and P8 to block the recruitment of Tregs into the neonatal skin from the thymus and lymph nodes. This mechanism has previously been demonstrated (10.1016/j.immuni.2015.10.016). We found that, similar to systemic Treg depletion, inhibition of local Treg recruitment led to reduced melanin abundance in hair cycle stage–matched follicles (New Figure 2E). These findings suggest that Tregs promote pigmentation through a local and skin-specific mechanism.

Reviewer #3:

We appreciate Reviewer 3 also highlighting positive attributes of our manuscript who states:

“The study provides an intriguing exploration into the immunological mechanisms involved in skin pigmentation”

“The link established between ETregs and PPARγ activity in MeSCs during postnatal development and the paralleled disruptions observed in the human skin disorder vitiligo provide a strong translational significance to the study.”

Comment 1:

“How are the specific activation markers identified to assess Tregs and their relevance to the study's context?”

Thank you for this question. The activation markers used in this study, including CD25, CTLA4, and ICOS, were selected based on prior literature demonstrating that neonatal Tregs express higher levels of these markers compared to their adult counterparts (see Scharschmidt et al., *Immunity*, 2015). While previous studies focused on a single time point, our analysis revealed dynamic changes in the expression of these markers across the early postnatal period. These observations motivated us to explore the functional consequences of temporally distinct Treg populations, leading to our comparison of early (ETreg) and late (LTreg) Treg depletion. We have now clarified this rationale in the revised manuscript. (see page 3, paragraph 3 of revised manuscript).

Comment 2:

“In the manuscript, Tregs are described as accumulating in the skin by postnatal day 13. What mechanisms are proposed for this accumulation, and is there evidence to suggest a selective recruitment or retention of Tregs in the skin?”

This is an important point. As demonstrated by Scharschmidt et al., Tregs seed the skin during the neonatal period through thymic egress and in a microbiota-dependent manner. Interestingly, the skin appears to selectively recruit Tregs, as evidenced by a rising proportion of Foxp3 positive cells among CD4 T cells. This suggests either preferential homing or local retention of Tregs over conventional T cells. We have now referenced this mechanism more explicitly in the revised text. (see page 3, paragraph 1 of revised manuscript).

Comment 3:

“The study mentions the influx of Tregs in skin during the early neonatal period; could this be due to local proliferation or migration from peripheral sites?”

The accumulation of Tregs in neonatal skin has been shown to result from thymic migration rather than local proliferation or recruitment from peripheral lymphoid tissues (Scharschmidt et al., 2015). Our study builds on these findings by examining the consequences of removing these thymically derived Tregs during early life.

Comment 4:

“How does the P3-P12 interval influence the functional status of Tregs in the skin?”

We appreciate this thoughtful question. However, Treg numbers in the skin are extremely low at these early time points, which makes it technically difficult to isolate enough cells for functional assays such as RNA sequencing or in vitro experiments. While understanding functional changes across this interval is indeed an important area of research, it falls outside the scope of the current study, which is focused on how the presence or absence of early neonatal Tregs shapes pigmentation through keratinocyte and melanocyte interactions.

Comment 5:

“Is there a possibility that the punctual depletion of ETregs at P6-P8 has immediate versus delayed effects on MeSCs?”

Yes, this is a key insight. Our data support a model in which ETreg depletion at P6 and P8 leads to immediate transcriptomic changes in melanocyte lineage cells by P9. These early changes then result in a delayed, phenotypic consequence; a significant reduction in pigmentation that becomes apparent by P28. We have clarified this temporal relationship more clearly in the revised Results and Discussion sections (see page 9, paragraph 3 of revised manuscript).

Comment 6:

“Could the study's findings about PPAR γ signaling activity be applicable to other stem cell niches in the skin, such as the HFSCs?”

Thank you for this interesting suggestion. Given that melanocyte stem cells and hair follicle stem cells (HFSCs) share a common niche within the hair follicle, it is highly plausible that other resident cell populations could also be influenced by changes in PPAR gamma signaling. In support of this, we have performed functional experiments using Krt14 Cre Pparg flox mice, which lack PPAR gamma in the hair follicle epithelium, including HFSCs (see Reviewer 1, comment 6 above). These mice show significantly reduced melanin abundance compared to Pparg sufficient controls, suggesting that epithelial intrinsic PPAR gamma activity impacts not only melanocytes but potentially other niche components such as HFSCs. Further studies will be needed to dissect the broader influence of this pathway on additional stem cell compartments.

Comment 7:

“The manuscript suggests that melanocyte function is regulated by Tregs independent of secondary mediators. Can authors provide more details on this regulatory mechanism?”

Our data support a model in which Tregs regulate pigmentation by modulating PPAR gamma signaling in the hair follicle epithelium. This is supported by our findings in Krt14 Cre Pparg flox mice, where loss of PPAR gamma in the hair follicle results in reduced pigmentation at P28, as shown by Fontana Masson staining (Figure 4L and 4M). To explore potential upstream mediators, we performed bulk RNA sequencing on sorted neonatal Tregs and integrated this with our skin single cell RNA sequencing data. Using ligand receptor interaction analysis through CellChat, we identified Il10 and Tnfsf9 as Treg-derived factors predicted to activate PPAR gamma target genes in hair follicle keratinocytes. In the absence of Tregs, we observe downregulation of these targets, including IGF1, which is a known melanogenic factor. We confirmed this functionally by culturing melanocytes with IGF1, which enhanced pigment production (New Figure 5J and 5K). Together, these data suggest that Tregs modulate epithelial PPAR gamma signaling, which in turn supports melanocyte function through factors such as IGF1. For figures, please refer to response to reviewer 1.

Comment 8:

“What are the implications of the transient loss of Tregs on systemic inflammation and how does it correlate with skin pigmentation?”

We appreciate the reviewer’s concern that systemic inflammation could contribute to the pigmentation phenotype. However, we did not observe major differences in weight or general health between control and ETreg depleted mice, suggesting the absence of overt systemic inflammation. Furthermore, the pigmentation phenotype is reproduced through epithelial specific deletion of PPAR gamma, supporting a local tissue intrinsic mechanism rather than a secondary consequence of systemic immune dysregulation.

Comment 9:

“Can authors further explain the importance of a specific ‘window’ of time for Treg requirement in skin pigmentation?”

This is a key concept in our study. We observed that depletion of early Tregs between P6 and P8 leads to significant downregulation of melanocyte specific genes and results in a lasting pigmentation defect. In contrast, depletion of Tregs at later time points (P10 to P12) does not affect pigmentation or melanocyte marker gene expression. These findings suggest that the presence of Tregs during a specific window in early postnatal life is essential for supporting melanocyte development and function, likely coinciding with a critical phase of hair follicle morphogenesis. Once this window has passed and melanocytes have matured, they may no longer require Treg mediated support.

Comment 10:

“How does the transcriptomic dysregulation of the PPARγ signaling pathway immediately post ETreg depletion manifest on a cellular level?”

Immediately following Treg depletion, we observed a reduction in PPAR gamma target gene expression within hair follicle epithelial cells, most notably IGF1. IGF1 is a known regulator of melanocyte activity and has been shown to enhance melanogenesis in both our in vitro experiments and prior published studies (<https://doi.org/10.1007/s11626-016-0052-y>). This early disruption in signaling may compromise melanocyte stimulation and contribute to the long term pigmentation defect observed. Beyond its role in melanogenesis, PPAR gamma activity is also critical for lipid metabolism and for suppressing skin inflammation (doi: 10.3390/ijms22168634), indicating that Treg driven modulation of this pathway may have broader implications for skin homeostasis.

Comment 11:

“What additional immune cell subsets could be influenced by ETreg depletion that were not assessed in this study?”

We thank the reviewer for this thoughtful question. While our flow cytometry focused primarily on T cell populations, we agree that other immune subsets, particularly myeloid cells, may also be affected by ETreg depletion. Although we captured myeloid populations via single cell RNA sequencing, we acknowledge that this technique is limited in its ability to efficiently detect certain immune subsets, such as neutrophils or rare dendritic cell populations. Importantly, the single cell RNA-seq dataset generated from this study provides a valuable and accessible resource for further exploration. We anticipate that both our group and others in the community will build on this dataset to investigate additional immune cell dynamics in the neonatal skin. We are currently planning follow-up studies to expand this analysis and more fully characterise the broader immunological landscape influenced by early-life Tregs.

Comment 12:

“Is there potential redundancy in the signaling pathways that regulate MeSC function, and could other pathways compensate for the loss of PPAR γ signaling?”

This is an important point. Other PPAR isoforms, including PPAR alpha, have been shown to regulate melanocyte pigmentation in vitro. However, in our study, we used isoform-specific PPAR gamma agonists and antagonists, and these alone were sufficient to modulate skin pigmentation. Moreover, previous work has shown that PPAR gamma, but not PPAR alpha or PPAR beta, exhibits peak expression during the postnatal development of skin (doi.org/10.1083/jcb.200011148). We have now expanded the discussion to make this point clearer and to explain why we believe PPAR gamma plays a dominant role in this setting (see page 12, paragraph 2 of revised manuscript).

Comment 13:

“How does the involvement of Tregs in the suppression of type 2 helper T cell mediated fibrous pathology contribute to skin pigmentation?”

We appreciate this connection. Boothby et al. (doi: 10.1038/s41586-021-04044-7) have shown that neonatal Tregs prevent a fibrotic pathology in the skin, which is associated with a transient loss of dermal white adipose tissue. It is therefore possible that this could indirectly influence pigmentation. However, the fibrotic phenotype they describe occurs at later time points (after P25, following Treg depletion at P8 and P15). In contrast, our model targets earlier time points, and we observe immediate transcriptomic changes in PPAR gamma signaling as early as P9, along with long-term effects on pigmentation by P28. Additionally, we do not observe signs of fibrotic pathology following early Treg depletion. These kinetic differences support the idea that Treg function is developmentally stage specific, and based on our current data, we believe the pigmentation defect described here is not linked to fibrotic remodeling.

Comment 14:

“What is the functional role of Tregs in supporting HFSC differentiation, and how does this interact with MeSC-mediated skin pigmentation?”

We thank the reviewer for raising this interesting point. Our lab has previously shown that Tregs can support HFSC proliferation and differentiation via expression of the Notch ligand Jag1 (doi: 10.1016/j.cell.2017.05.002). Since hair follicle epithelial cells produce C-kit ligand, which is critical for melanocyte survival, it is plausible that Treg interactions with HFSCs indirectly support MeSC function. However, in this study, we did not observe defects in HFSC number or proliferation following ETreg depletion. While the relationship between HFSC differentiation and MeSC function is intriguing, it falls outside the scope of the current work.

Comment 15:

“Can the results observed in the mouse model regarding ETreg depletion and the PPAR γ pathway be translated to human skin development or skin pigmentation disorders directly?”

We believe there is translational relevance to our findings. PPAR gamma agonists have been shown to promote pigmentation in human skin (doi: 10.1111/j.1600-0625.2006.00521.x). Moreover, genome-wide association studies in vitiligo have identified variants in genes associated with Treg biology, including Foxp3, Ctla4, and Il2ra (doi: 10.1016/j.det.2016.11.013). In our own analysis of human vitiligo scRNA-seq data, we observe reduced signatures of PPAR gamma activity in lesional skin. While the underlying mechanisms may differ between mouse and human, these results suggest a potential role for Treg mediated modulation of PPAR gamma signaling in human pigmentation disorders.

Comment 16:

“The manuscript posits that the PPAR γ pathway is critical for melanocyte function. Are there other pathways with potential involvement in this process that warrant further investigation?”

Yes, other pathways may contribute to melanocyte regulation. For example, PPAR alpha has also been shown to modulate pigmentation in vitro. In addition, signaling pathways such as TGF beta and Wnt are known to influence melanocyte biology more broadly. However, in our dataset, we did not observe transcriptomic changes consistent with perturbation of these pathways following Treg depletion. We have now clarified this point in the revised Discussion to acknowledge that while additional pathways may play roles, our current data support a primary role for PPAR gamma signaling in this model. (see page 22, paragraph 3 of revised manuscript).

Comment 17:

“Given that ETregs were found to be essential for HF PPAR γ signaling, what are the broader implications for the understanding of immune cell involvement in tissue development?”

This is an important and exciting area. A growing body of literature supports the idea that immune cells can shape tissue development through interactions with local stem or progenitor populations. For instance, macrophages have been shown to regulate stem cell maintenance in the hair follicle and other tissues. Our findings contribute to this emerging view by demonstrating that Tregs can influence epithelial signaling in a non-inflammatory context, with consequences for pigment-producing cells. These insights may help refine our broader understanding of immune-epithelial cross-talk during postnatal development.

Comment 18:

“How would the findings of this study influence therapeutic approaches to skin pigmentation disorders such as vitiligo?”

Vitiligo is primarily driven by CD8 positive T cell mediated destruction of melanocytes. Our findings suggest that keratinocyte intrinsic PPAR gamma activity not only supports melanocyte function but may also contribute to suppression of local inflammation. Indeed, decreased PPAR gamma signaling has been observed in vitiligo lesional skin, and PPAR gamma agonists have been shown to enhance melanogenesis in vitro. These observations suggest that restoring PPAR gamma signaling in the epidermis could serve a dual purpose in vitiligo: promoting melanocyte function and limiting cytotoxic T cell accumulation.

Comment 19:

“The manuscript reports a reduction in MeSC proliferation upon ETreg depletion. Could this be linked to specific changes in the cell cycle or apoptotic pathways?”

We thank the reviewer for pointing this out. To clarify, the MeSC proliferation phenotype was observed in experiments involving a full four-dose depletion of Tregs, not the standard two-dose early depletion used in most of our analyses. We have now clarified this in the text to avoid confusion. (see page 5, paragraph 1-2 of revised manuscript). Further investigation would be required to determine whether this reduced proliferation is linked to changes in cell cycle dynamics or apoptotic signaling.

Comment 20:

“How does the heterogeneity of Tregs, possibly containing subsets with distinct roles and activation states, affect the overall interpretation of the results?”

We agree that Treg heterogeneity is an important and emerging topic. However, this question lies outside the scope of the present study. Due to the limited number of Tregs present in neonatal skin, it is technically very challenging to perform subset-level analyses or functional validation in this context. Moreover, the heterogeneity of neonatal skin Tregs has not been well characterized in the literature to date. While this is a compelling direction for future research, our current findings focus on the functional requirement for the Treg population as a whole during a critical developmental window.

Comment 21:

“Can authors discuss the significance of the timing of ETreg depletion on P6-P8 and how this differs from potential interventions at other developmental stages?”

This is an important point. Melanocytes are known to seed the hair follicle niche early in development, but the period during which they acquire full functional competence is less well defined. Our observation that depletion of early Tregs between P6 and P8 leads to long-term pigmentation defects, while depletion between P10 and P12 does not, suggests that the P6 to P8 window may represent a critical period of melanocyte functional specification. We have now clarified the developmental rationale for this timing in the revised Discussion. (see page 21, paragraph 2 of revised manuscript).

Comment 22:

“The authors discussed the role of ETregs in melanocyte stem cell function. Can authors provide more details on the experimental evidence supporting a direct interaction between these cells?”

Due to the extreme scarcity of Tregs in neonatal skin, it is technically challenging to perform direct interaction studies between Tregs and melanocytes in vivo or ex vivo. In this study, we relied on integrative single cell RNA sequencing analysis, which suggested that Tregs influence keratinocyte gene expression. In turn, these keratinocytes regulate melanocyte activity through factors such as IGF1. While this supports an indirect mode of regulation, it does not rule out the possibility of direct Treg melanocyte interactions, which remains an interesting avenue for future exploration.

Comment 23:

“How does the study address the potential compensatory mechanisms that may arise from the depletion of ETregs?”

Thank you for this thoughtful question. Neonatal Tregs have been shown to exhibit stronger suppressive function compared to adult or peripheral Tregs (see Mathis and Scharschmidt, see references: 10.1126/science.aaa7017, and 10.1016/j.immuni.2015.10.016). Following early Treg depletion in our model, we observed a robust compensatory influx of Tregs into the skin, leading to higher Treg numbers in Δ ETreg mice than in controls by P28. We hypothesise that this replenishment reflects an attempt to restore immune balance, but that the incoming Tregs may differ in function or origin. The phenotypic consequences of this compensatory response were not explored in detail in the current study, but we

acknowledge it as a relevant consideration in the Discussion. (see page 21, paragraph 2 of revised manuscript).

Comment 24:

“How does the study address the heterogeneity within Treg and MeSC populations, and could there be subpopulations with distinct roles?”

This is an insightful point. Due to the limited number of Tregs present in neonatal skin, our study was not designed to resolve their heterogeneity. Investigating Treg subsets in this setting remains technically challenging and is an area for future investigation. Regarding MeSCs, prior work in adult skin has identified long-lived and short-lived subtypes (see Ito et al.). Whether similar subtypes exist during early postnatal development is currently unknown and falls beyond the scope of this manuscript.

Comment 25:

“In Figures 3B and 2C, the color legends “-log(10)FDR” should be corrected as “-log₁₀(FDR)”.”

We thank the reviewer for spotting this. We have now corrected the figure legends to use the appropriate notation.

Comment 26:

“It is unclear how the differentially expressed genes (DEG) were identified. Which test/statistical model was used? Is it comparing one cell type and the pool of all the remaining cell types, or is it comparing any pair of cell types and then aggregate in some way? Can authors explain why some cell types do not show any DEGs?”

We thank the reviewer for this important question. DEGs were identified using the FindMarkers function in Seurat. For each individual cell type, we performed differential expression analysis between control and Δ ETreg conditions. This within-cell-type comparison avoids confounding by comparing like populations across experimental groups. The statistical model used was the Wilcoxon Rank Sum test, which is the default method in Seurat. To ensure robustness, we applied multiple testing correction using either Bonferroni or Benjamini-Hochberg methods. Cell types that did not yield statistically significant DEGs after adjustment (adjusted p-value greater than 0.05) were not reported. This conservative threshold avoids overinterpreting noise in sparsely represented cell populations.

Comment 27:

“In Figures 5K and 6E, the gene expression levels range from -3000 to more than 2000? Did authors normalize the data by sequencing depth and perform any reasonable transformation? This will affect the robustness of the single-cell RNA-seq analysis.”

We appreciate the reviewer’s close attention to these data visualizations and welcome the opportunity to clarify our preprocessing pipeline.

1. **Normalization and scaling:** All single cell RNA-seq data were normalized for sequencing depth using Seurat’s NormalizeData function, which applies log-normalization. This corrects for differences in sequencing depth across cells.
2. **Figure 5K (heatmap):** This figure shows scaled expression values (mean-centered and variance-scaled). Depending on the data layer used (data, counts, or scale.data), Seurat may exponentiate values for averaging or apply further transformations before visualization. This may result in large numerical ranges that reflect transformed relative expression rather than raw counts.
3. **Figure 6E (module score):** This panel uses Seurat’s AddModuleScore function to calculate PPAR gamma activity scores for individual cells based on a predefined gene set. The score represents the average expression of the gene set relative to randomly selected background genes. Values can be negative or positive depending on whether the gene set is under- or over-expressed relative to background.

In summary, all values presented reflect normalized and transformed data. The observed ranges are not raw counts but derived scores or scaled expressions appropriate for their respective visualization strategies.

Comment 28:

“Authors stated that “Neonatal Tregs regulate the hair follicle transcriptome and PPAR γ activity”. What evidence supports the causality between neonatal Treg presence and changes in the hair follicle transcriptome, rather than a correlation or secondary consequence of other developmental processes?”

Causality is supported by our depletion model, in which Tregs are selectively ablated between P6 and P8. When compared to Treg-sufficient controls, Δ ETreg skin shows clear changes in the hair follicle transcriptome as early as P9, including reductions in PPAR gamma targets. The timing and specificity of this depletion provide strong evidence that the observed changes are a direct consequence of Treg absence, rather than a secondary effect of broader developmental variation.

Comment 29:

“How do authors rule out the possibility that observed transcriptomic changes in the hair follicle are independent of Treg activity and could be attributed to the intrinsic maturation of the hair follicle itself?”

We appreciate this point and agree that intrinsic hair follicle maturation could be a confounding factor. Using previously published scRNA-seq datasets (by Driskell lab, available to explore at <https://skinregeneration.org/>), we explored the expression of PPAR γ target genes (Fabp7, Adipoq, Abca1, Apoe, Cd36) that we have identified as being under Treg control. These genes were minimal correlation with the progression of development between postnatal day 0, 21 and 49. This supports the interpretation that the transcriptomic changes are not simply due to delayed development but are associated with the loss of Treg function. Furthermore, both single cell and bulk transcriptomic profiling of the skin were performed immediately after depletion of ETregs, minimising the possibility that delayed developmental timing is responsible for the transcriptomic changes in the hair follicles after depletion of ETregs.

Comment 30:

“Can the study distinguish between direct effects of Tregs on hair follicles versus indirect effects mediated by other cell types or signaling molecules in the skin microenvironment?”

We have addressed this question in part through computational ligand-receptor interaction analysis. Using bulk RNA-seq from sorted Tregs (from skin and skin-draining lymph nodes on P9), we performed NicheNet analysis to identify potential downstream targets of Treg-derived ligands in the hair follicle epithelium. This analysis suggests that Tregs in the skin produce IL10 and TNFSF9, which may regulate the expression of Apoe, Pltp, and Abca1 in keratinocytes. These are known regulators of PPAR gamma signaling and lipid metabolism. While we cannot exclude additional indirect effects, this in silico evidence supports a model in which Tregs directly influence hair follicle transcriptional programs through well-defined ligand-receptor pathways. These findings are now described more clearly in the revised Results and Discussion sections.

Comment 31:

“What methodologies were used to exclude the influence of global developmental changes occurring in neonatal skin that might confound the specific attribution to Tregs?”

We acknowledge that fully disentangling developmental and Treg-mediated effects in vivo is technically challenging. To minimise variability, all experiments were performed using littermate controls, which ensures that animals are matched for age and developmental stage. We observed no notable differences in hair follicle morphology or maturity at P9 between control and Δ ETreg skin, suggesting that the observed transcriptomic changes are not due to differences in overall developmental timing. Instead, the consistent transcriptional shift in the Δ ETreg group supports a specific effect of Treg absence.

Comment 32:

“How were alternative pathways or transcription factors that might regulate the PPAR γ pathway in hair follicles independently of Tregs accounted for in the analysis?”

Our study was designed to investigate the role of Tregs in modulating hair follicle PPAR gamma activity. While we acknowledge that other pathways and transcription factors may contribute to PPAR gamma regulation, investigating Treg-independent mechanisms lies outside the scope of this manuscript.

Comment 33:

“Were the findings validated in vitro, such as in hair follicle organ cultures, where the Treg influence can be modulated independently of the in vivo environment?”

We now include in vitro data demonstrating that culturing Melan A melanocyte cells in the presence of IGF1 enhances pigmentation. IGF1 is a known PPAR gamma target and was found to be downregulated in the hair follicle compartment following early Treg depletion. These findings support our in vivo results and highlight a mechanistic link between Treg-mediated PPAR gamma signaling and melanocyte activation.

Comment 34:

“In the single-cell analysis, how was the specificity of PPAR γ activity scores to Treg interactions ascertained against the background of the complex skin cellular milieu?”

We appreciate this nuanced question. Given the complexity of the tissue microenvironment and the difficulty of defining direct Treg-cell interactions at single cell resolution, we did not attempt to assign PPAR gamma activity changes solely to direct Treg contact. Instead, we used CellChat analysis to identify likely ligand-receptor interactions. This approach highlighted IL10 and TNFSF9 as Treg-derived factors that may influence the expression of PPAR gamma target genes in the hair follicle. Although PPAR gamma can also be regulated by lipid-derived ligands, Tregs are not known to be a major source of these molecules. Therefore, we interpret the observed changes in PPAR gamma activity scores in the context of altered paracrine signaling in the absence of Tregs.

Comment 35:

“Are there data to show that manipulation of Treg levels or function does not alter the developmental timing of hair follicle maturation, which could indirectly affect transcriptomic profiles?”

We agree that a direct correlation between Treg function and the timing of hair follicle maturation in neonatal skin has not been comprehensively defined. Our data show that Treg numbers increase over early postnatal development, raising the possibility that they could influence aspects of follicle maturation. However, as discussed in our response to Comment 29, the differentially expressed genes observed in the hair follicle following ETreg depletion are not strongly associated with canonical hair follicle developmental programs. Together, this suggests that the transcriptomic changes we observe are more likely to reflect regulatory rather than maturational effects.

Comment 36:

“What are the temporal aspects of the Treg impact on hair follicle transcriptomics? Is this a transient phase during neonatal development, or are there long-lasting effects?”

By seven weeks of age, we no longer observe overt morphological differences in the hair follicles between control and Δ ETreg mice, suggesting that the impact of early Treg depletion on follicle structure is transient. However, prior studies have demonstrated that epithelial cells can retain long-lasting epigenetic changes following inflammatory or immune perturbations, including Treg loss (e.g., 10.1038/nature24271). Although we do not directly assess chromatin accessibility or epigenetic remodelling in this study, our findings raise the possibility that early Treg-dependent signalling may imprint lasting transcriptional or epigenetic memory in the hair follicle niche.

As a final point, we would like to emphasise that in this study we uncover a previously unappreciated role for early postnatal regulatory T cells in regulating melanocyte activity through modulation of PPAR gamma signalling in the hair follicle. This work provides novel insights into immune-stem cell crosstalk during tissue development and presents broader implications for understanding immune-mediated control of epithelial homeostasis. With the addition of new experiments and analyses in response to the reviewers' thoughtful suggestions, we hope the reviewers and editors will agree that the revised manuscript is substantially strengthened and now fully addresses the points raised.

Sincerely,

Niwa Ali, PhD
Group Leader, Senior Lecturer in Cellular Immunology & Wellcome Trust Sir Henry Dale Fellow

Corresponding author:

Dr Niwa Ali. Wellcome Trust Sir Henry Dale Fellow, Immune-epithelial cell interactions Laboratory.
E-mail: niwa.ali@kcl.ac.uk

Affiliation for corresponding author:

Centre for Gene Therapy and Regenerative Medicine, School of Basic and Biomedical Sciences,
Guy's Hospital, Tower Wing 28th Floor, King's College London, London SE1 9RT, UK.

Table. Top 200 Differentially expressed genes in melanocytes from control and Δ ETreg neonatal mouse skin (unadjusted p-values).

Gene.name	avg_log2FC	p_val
Cma1	-5.1560536	4.35E-06
Atox1	-1.2483728	2.42E-05
Naa38	-1.4211993	7.56E-05
Prrc2c	1.3633841	0.00070687
Tmsb10	-0.8667854	0.00090306
VeZF1	0.96672392	0.00094713
Ogn	-1.7777865	0.00103577
Dmxl1	1.0852879	0.0012506
Map1b	1.2880659	0.00135823
Sec61g	-1.0335837	0.00138357
Fbn1	-1.2957775	0.00159667
Srsf7	1.13418566	0.00162464
Lox	-1.4960916	0.00171911
Xist	-1.302544	0.00187625
Dpm3	-0.9860053	0.00216533
Fam129b	1.15529022	0.00222181

Lum	-2.0408253	0.00240312
Srgn	-2.0226091	0.00240312
Tpsb2	-3.7384984	0.00245033
Gpc3	-2.3660934	0.00250995
Sec61b	-0.8541785	0.00259407
Rpl38	-0.7021512	0.00299776
Atrn	0.96368179	0.00321546
Max	0.82608113	0.00321546
Elob	-0.6517933	0.0034952
Ndufb1-ps	-0.6266982	0.00368758
S100a11	-0.9103904	0.0044501
Rock1	1.02567113	0.00487907
Brd4	1.05783113	0.00505712
Phactr2	1.15399655	0.00505712
Dact1	1.16220222	0.00510527
Pfn1	-0.5484576	0.00520633
Hnrnp1	0.8709305	0.00530989
Map4k4	1.10967811	0.00531905
Aqp1	1.02531623	0.00533924
Cyp11a1	-1.7677034	0.00551958
Dcp2	-0.645893	0.00551958
Egfr	-0.9431165	0.00551958
Ids	-0.6797599	0.00551958
Ltbp1	-0.4322989	0.00551958
Mdfic	-1.2682222	0.00551958
Ntn1	-1.4538347	0.00551958
Parp4	-0.5292489	0.00551958
Rgs18	-1.5516457	0.00551958
Rpp25l	-0.7872982	0.00551958
Tyrobp	-1.4624438	0.00551958
Capza1	-1.1331685	0.0058608
Csnk1e	-1.1752774	0.00611779
Cdc37l1	1.07831915	0.00637838
Rps27	-0.4713688	0.00658461
Tm4sf1	1.25920547	0.00667502
Lbh	0.96060987	0.00667692
Uba52	-0.7825741	0.00667803
Tanc2	1.00088959	0.00678149
Nsd2	-0.769497	0.00778972
Ankrd10	0.80039671	0.00785997
Ccnt1	0.87925934	0.00785997
Fbxw7	0.77524243	0.00785997
Gnptg	0.99711761	0.00785997

Nemf	0.75675453	0.00785997
Islr	-1.1306253	0.00799285
AY036118	1.12546371	0.00838399
Hadhb	0.9885995	0.00863269
March2	0.90299593	0.00900445
Dars	-0.8179241	0.00930762
P4hb	-0.7452508	0.00946496
Nr4a3	-1.7127934	0.00960892
Mtdh	0.86278494	0.00961585
Fn1	-2.3201523	0.00972714
Rps20	-0.396137	0.00978359
St3gal4	1.15237103	0.00989384
Rpl35	-0.4364707	0.01026818
Ppp6r1	-0.5573696	0.01052141
Top1	0.97724311	0.01072791
Purb	0.77446427	0.01135089
Net1	-0.5885227	0.01151019
Tmf1	0.76098162	0.011552
Plp1	0.82838359	0.0117406
Cdc42bpa	-1.0465777	0.01203356
4833420G17Rik	0.73639864	0.01208298
Ccny	0.65219793	0.01208298
Fam193a	0.62338101	0.01208298
Pdzd8	0.68740058	0.01208298
Prpf39	0.67860607	0.01208298
Stk25	0.63660186	0.01208298
Tbc1d23	1.02770677	0.01208298
Pigt	-0.6872528	0.01222929
Fdps	-0.8119063	0.01258057
Nbl1	-0.8278701	0.01322051
Snhg18	-0.7688598	0.01359654
Safb2	0.69729983	0.01491495
Sorbs1	0.99040741	0.01529552
Gng5	-0.6762644	0.01554307
Ccnt2	0.81514865	0.01604814
Eif2s2	-0.7993456	0.01655366
Sf3b1	-0.6924248	0.01655366
Abca5	-0.7318808	0.01748101
Adora3	-1.0398126	0.01748101
Arhgef6	-0.7731524	0.01748101
Cfh	-1.4722168	0.01748101
Clec12b	-0.9395782	0.01748101
Dhdds	-0.3283118	0.01748101

Dpt	-1.5648598	0.01748101
Fam117a	-0.5758443	0.01748101
Gas8	-0.7279074	0.01748101
Gata2	-1.4548544	0.01748101
Gemin2	-0.4600223	0.01748101
Gpr65	-0.952994	0.01748101
Haus7	-0.3560169	0.01748101
Hist1h4i	-0.4010284	0.01748101
Hmces	-0.3774513	0.01748101
Hs3st1	-0.9073159	0.01748101
Igf2	-1.1275388	0.01748101
Il1rl1	-1.1585461	0.01748101
Il4	-1.0793268	0.01748101
Inpp5b	-0.8138796	0.01748101
Lrch2	-0.5407072	0.01748101
Mrgprb1	-1.4283377	0.01748101
Mrgprb2	-1.4559274	0.01748101
Msrb3	-0.6097984	0.01748101
Naa30	-0.2686643	0.01748101
Optn	-0.7314759	0.01748101
Pigs	-0.5521785	0.01748101
Pih1d1	-0.747818	0.01748101
Plek	-1.1419179	0.01748101
Polr3b	-0.4125293	0.01748101
Rgs1	-1.308942	0.01748101
Ruvbl1	-0.6116613	0.01748101
Sept6	-0.7730769	0.01748101
Setdb2	-0.6857599	0.01748101
St6galnac4	-0.5110984	0.01748101
Svip	-0.6160385	0.01748101
Taok3	-0.8418803	0.01748101
Tesk2	-0.8503227	0.01748101
Utp15	-0.3824854	0.01748101
Vps13d	-0.3625233	0.01748101
Zbtb5	-0.2809349	0.01748101
Zfp763	-0.4653951	0.01748101
Zfp992	-0.8371381	0.01748101
Hip1	1.05513185	0.01809147
Lamtor4	0.81385448	0.01835428
Megf10	0.77287463	0.01835428
Calhm2	0.68336163	0.01839001
Cep170	0.72944363	0.01839001
Chpf2	0.71599206	0.01839001

Decr1	0.68327748	0.01839001
Dleu2	0.85062849	0.01839001
Ep400	0.67629067	0.01839001
Fam89a	0.53998679	0.01839001
Ggnbp2	0.73018488	0.01839001
Lpgat1	0.68289877	0.01839001
Numb1	0.91391266	0.01839001
Nup93	0.68509086	0.01839001
Ppp6c	0.73911593	0.01839001
Rbm27	0.61432645	0.01839001
Zfp282	0.55452881	0.01839001
Atp5e	-0.5715636	0.01880143
Uqcr10	-0.822916	0.01892051
Cd59a	0.69003275	0.01919126
Snx2	0.65414496	0.01954572
Hsd17b11	1.19725886	0.02030418
Ptprz1	0.78835453	0.02053505
Rps3a1	0.53196848	0.02056643
Rps9	0.49001782	0.02056643
Tmpo	-0.6306965	0.02092677
Jmjd1c	0.82699853	0.02102
Strn4	0.97333876	0.02117261
Rasa3	0.74367518	0.02119577
Eef1a1	0.48372102	0.02148912
Brd9	0.78899133	0.02156593
Gbp7	0.77832311	0.02156593
Yme1l1	0.92306226	0.02156593
Snf8	-0.7305502	0.02222203
Col1a2	-1.5965573	0.02259862
Il11ra1	-1.0803262	0.02268833
Ube2j1	-0.8491897	0.02268833
Fktn	0.76048716	0.02289541
Rad21	0.69164304	0.02289541
Etv1	1.34998773	0.02321064
Usp19	0.65945296	0.02330886
Ptma	0.41185172	0.02347051
Ralgapa1	0.8356148	0.02358571
Cep85l	0.80099584	0.02362687
Cox7a2	-0.7764608	0.02390774
Mrpl3	-0.726328	0.02393748
Atxn2	0.98998864	0.02430893
Thoc2	0.79421691	0.02430893
Akap13	1.15685617	0.0243318

Ythdc1	0.65158076	0.02493068
Acsl4	0.96347247	0.02506806
Emsy	0.62849222	0.02506806
Slirp	-0.8525298	0.02560564
Hdc	-2.1867966	0.02562064
Ms4a2	-1.4729583	0.02562064
Nt5e	-0.7670667	0.02562064
Pofut1	-0.7556948	0.02562064
Slc25a32	-0.9872626	0.02562064
Tnfaip3	-1.2762297	0.02562064
Uap111	-0.8166326	0.02562064
Btg3	-0.822384	0.02582689

Re: Manuscript # NCOMMS-24-09213,
" Early skin seeding regulatory T cells modulate PPAR γ -dependent skin pigmentation"

Dear Editors,

We thank the reviewers for their valuable time and thoughtful feedback, and for recognising the substantial new data and revisions that were incorporated into our revised manuscript.

Response to Reviewer 1

Reviewer #1:

In this revision, the authors provided a large amount of new data, including new RNA-seq experiments showing the molecular links between ETreg, HFSC, and MeSC, the keratinocyte-specific PPAR γ conditional knockout mouse showing PPAR γ 's unique role in HFSCs. The text of the manuscript has been revised extensively to be more accurate and clearer, addressing the reviewer's previous concerns. These changes have significantly elevated the quality of the manuscript. The reviewer shares the excitement of the authors' surprising findings that established a novel link between neonatal skin Treg cells and MeSC-mediated skin pigmentation.

Author Response:

We sincerely thank the reviewer for their kind words and encouragement. We are pleased that the additional data and revisions have addressed the earlier concerns and that our findings were met with enthusiasm.

Response to Reviewer 2

Reviewer #2:

Thank you for your detailed responses to the review comments. Overall, your revisions effectively address the points raised. The key supplemental experiments resolve the primary concern regarding potential hair cycle confounding of the pigmentation phenotype, and the new sequencing data provide additional support for the role of keratinocyte-intrinsic PPAR γ signaling in regulating melanocyte activity. I acknowledge the technical challenges encountered with DCT staining and support the removal of suboptimal immunofluorescence images to enhance data clarity. For future melanocyte visualization studies, SOX10 or S100 could be considered as alternative markers. Note that these also label Schwann cells, but distinction may be achievable through location-specific assessment combined with morphological evaluation. The manuscript shows clear improvement in its revised form.

Author Response:

Thank you for this generous and constructive feedback. We are grateful for your support regarding the removal of the suboptimal histological data and appreciate your helpful suggestions about potential alternative markers for future melanocyte visualisation studies. We will certainly keep SOX10 and S100 in mind for future work.

Response to Reviewer 3

We thank the reviewer for their continued feedback. Please find our point-by-point responses below.

Comment 11:

Can authors incorporate the discussion on limitations into the Discussion section?

Author Response:

We have now incorporated this into the Discussion section. The following sentence has been added to acknowledge the limitations of our single-cell and flow cytometric profiling:

“Other cell types, such as neutrophils, may contribute to the regulation of PPAR γ activity following ETreg depletion. However, these cell types remain unaddressed by our flow cytometric profiling and single-cell RNA-seq analysis, which are unable to capture these populations.”

Comment 26:

Can authors provide full details of the methods used in the Methods section to ensure reproducibility?

Author Response:

We have now updated the Methods section with the requested details.

“Analysis was performed using Seurat package. Differential gene expression analysis was performed using FindMarkers function, which uses the Wilcoxon Rank Sum test followed by Bonferroni multiple test correction. Statistical significance was inferred by adjusted p-value of less than 0.05.”

Comment 27:

For Figures 5K and 6E, please clearly indicate that the values are scaled for visualization purposes only. Also, describe the scaling or transformation procedures used; otherwise, the interpretation of the values may be misleading.

Author Response:

Figures 5K and 6E have been updated to indicate that the values are scaled for visualisation purposes..

Comment 29:

Please consider including the results of this analysis in the Supplementary Materials.

Author Response:

We understand the reviewer's interest in including the exploratory comparison to the Diskell lab dataset. However, as noted in our prior response, this was an informal analysis using a public browser for exploratory purposes, and no formal or reproducible dataset was generated. Given the lack of quantitative metrics and the exploratory nature of the comparison, we do not believe it is suitable to include this in the Supplementary Materials. Instead, we have chosen to describe the rationale in our response for transparency while limiting the supplementary content to formally generated data.

We are grateful for the reviewers' feedback and hope that the revised manuscript is now fully acceptable for publication.

Sincerely,

Niwa Ali, PhD
Group Leader, Senior Lecturer in Cellular Immunology & Wellcome Trust Sir Henry Dale Fellow

Corresponding author:

Dr Niwa Ali. Immune-epithelial cell interactions Laboratory.
E-mail: niwa.ali@kcl.ac.uk

Affiliation for corresponding author:

Centre for Gene Therapy and Regenerative Medicine, School of Basic and Biomedical Sciences, Guy's Hospital, Tower Wing 28th Floor, King's College London, London SE1 9RT, UK.